# Differential nucleosome organization in human interphase and metaphase chromosomes

Keren Li [1,2,3,6], Irem Unlu [4,6], Yiren Tu [4,6], Lilien N Voong [4], Yanyan Lu [2,4], Brody Kendall[1], Xiaotian Ma[1], Sin Lei Pui[4], Meng Tao[4], Ji-Ping Wang [1,2,5 ✉] & Xiaozhong Wang [2,4,5 ✉]

## Abstract

**DNA bendability plays a critical role in stabilizing nucleosome assembly, yet its contribution to nucleosome dynamics in vivo remains poorly understood. Here, we applied chemical mapping to generate high-resolution nucleosome positioning maps at single-base-pair resolution from human interphase and metaphase chromosomes, revealing distinct patterns of nucleosome organization between the two states. Notably, we observed a unifying pattern of nucleosome positioning near euchromatic landmarks, including promoters, enhancers, and insulators, during mitosis. Interphase nucleosomes exhibited extensive repositioning, marked by increased nucleosome density, reduced spacing between nucleosomes, and the appearance of additional fragile nucleosomes compared to metaphase. Furthermore, our results show that metaphase nucleosomes display significantly higher DNA cyclizability around the dyad axis, whereas interphase nucleosomes, particularly those near regulatory regions, tend to position DNA with greater cyclizability at the edges of the nucleosome. Together, these findings highlight a dynamic interplay between DNA mechanics and nucleosome organization during the cell cycle.**

**Keywords** Chemical Mapping; Mitosis; DNA Cyclizability; Chromosome; Nucleosome
**Subject Category** Chromatin, Transcription & Genomics

## Introduction

The nucleosome, consisting of DNA wrapped around histone proteins, is fundamental to chromatin structure and gene expression (Kornberg and Lorch, 2020; Misteli, 2020). Structural and biophysical studies show that in canonical nucleosomes, histones primarily interact with the phosphodiester backbone of DNA rather than forming base-specific contacts (Luger et al, 1997). Instead, DNA sequence contributes to nucleosome positioning by modulating the mechanical propensity to form the superhelical conformation required for nucleosome assembly (Andrews and Luger, 2011; Struhl and Segal, 2013; Widom, 2001). Among these interactions, the contact between the nucleosomal dyad and the histone H3–H4 tetramer plays a central role in governing nucleosome stability (Brower-Toland et al, 2002; Luger et al, 1997; Thåström et al, 2004). It is important to note that most biophysical insights into sequence-dependent nucleosome positioning have been derived from in vitro studies using a limited number of sequences, such as the "601" and "5S rRNA". Two recent technical advances have expanded these investigations into cellular contexts at the genomic scale. First, a chemical biology approach was developed for high-resolution nucleosome mapping in vivo (Brogaard et al, 2012; Voong et al, 2016). This method introduces a serine-to-cysteine substitution at position 47 of histone H4, enabling targeted hydroxyl radical-mediated cleavage of nucleosomal DNA near the dyad axis (Flaus et al, 1996). To date, this strategy has successfully produced single-base-pair resolution nucleosome maps in interphase yeast and mouse embryonic stem cells (Brogaard et al, 2012; Chereji et al, 2018; Moyle-Heyrman et al, 2013; Ramachandran et al, 2015; Voong et al, 2016). Second, in 2021, the Ha lab introduced Loop-seq, a high-throughput method that quantifies DNA cyclizability, which serves as a proxy for intrinsic bendability, across thousands of DNA sequences (Basu et al, 2021). When paired with chemically determined nucleosome positions, these measurements revealed that DNA bendability strongly influences in vivo nucleosome positioning in yeast and mouse cells (Basu et al, 2022; Basu et al, 2021). Leveraging Loop-seq experimental data, we developed DNAcycP and DNAcycP2, deep learning-based computational tools that accurately predict intrinsic DNA cyclizability from sequence (Kendall et al, 2025; Li et al, 2022). These tools now enable detailed exploration of the relationship between DNA mechanics and nucleosome organization in any system where high-resolution nucleosome maps are available.

Beyond well-studied canonical nucleosomes, increasing evidence indicates that nucleosomes can adopt diverse structural states (Brahma and Henikoff, 2020; Zhou et al, 2019). Nucleosome remodelers and transcription factors (TFs) are known to alter nucleosome positioning during various biological processes, like the cell cycle (Ahmad et al, 2024; Bulyk et al, 2023; Markert and Luger, 2021). During mitosis, chromosomes undergo drastic conformational changes at both global and local levels. Globally, metaphase

---

[1]Department of Statistics and Data Science, Northwestern University, Evanston, IL 60208, USA. [2]NSF-Simons Center for Quantitative Biology, Northwestern University, Evanston, IL 60208, USA. [3]Department of Mathematics, University of Alabama at Birmingham, Birmingham, AL 35294, USA. [4]Department of Molecular Biosciences, Northwestern University, Evanston, IL 60208, USA. [5]NSF-Simons National Institute for Theory and Mathematics in Biology, Chicago, IL 60611, USA. [6]These authors contributed equally: Keren Li, Irem Unlu, Yiren Tu. ✉E-mail: jzwang@northwestern.edu; awang@northwestern.edu

chromosome condensation involves the formation and compaction of mitosis-specific chromatin loops along a spiral axis, accompanied by the disassembly of interphase enhancer-promoter loops, topologically associating domains (TADs), and compartments (Gibcus et al, 2018; Naumova et al, 2013; Zhang et al, 2019). Locally, studies have shown that many nuclease-sensitive sites are preserved from interphase into mitosis (Hsiung et al, 2015; Oomen et al, 2019; Owens et al, 2019; Teves et al, 2016), with some nucleosome repositioning near nucleosome-depleted regions (NDRs) (Javasky et al, 2018; Kelly et al, 2010; Liang et al, 2015; Martínez-Balbás et al, 1995; Oomen et al, 2019; Owens et al, 2019). However, the precise details of nucleosome reorganization during mitosis remain unclear, largely due to the limited resolution of traditional nucleosome mapping techniques. High-resolution nucleosome positioning data are essential for identifying sequence-dependent features such as the 10-bp periodicity of dinucleotides, a signature of nucleosome-forming sequences (Brogaard et al, 2012; Moyle-Heyrman et al, 2013; Voong et al, 2016). Thus, a key unanswered question is how mitosis influences nucleosome positioning and to what extent intrinsic DNA mechanics contribute to the dynamic repositioning of nucleosomes during this crucial cellular transition.

Here, we applied a chemical biology approach to investigate nucleosome organization in the human genome during mitosis. We generated single-base pair resolution nucleosome positioning maps for interphase and metaphase chromosomes. Our data reveal distinct organizational patterns between these two states. During mitosis, nucleosomes undergo widespread repositioning at regulatory regions. Compared to metaphase, interphase chromatin displays pronounced shifts, including shortened linker lengths, the presence of fragile nucleosomes, and increased nucleosome density, all of which are features associated with active transcription following cell division. Using DNAcycP2 (Kendall et al, 2025), we further show that the intrinsic cyclizability of DNA plays a critical role in shaping this dynamic nucleosome landscape.

## Results

### The first chemical map of nucleosome positioning for the human genome

The human genome encodes 15 histone H4 genes, each with a distinct DNA sequence but producing an identical protein. To replace the majority of endogenous H4 proteins with a mutant variant (H4S47C), we developed a two-step strategy. First, we identified a short hairpin RNA (shRNA) targeting a conserved region shared by all H4 genes (Fig. EV1A). Second, we used a PiggyBac (PB) transgenic system to express the H4-targeting shRNA and an RNAi-resistant H4S47C transgene simultaneously in HeLa S3 cells (Voong et al, 2016), thereby mitigating the deleterious effects of H4 depletion (Fig. EV1B,C). Clones exhibiting normal growth and morphology were selected for chemical cleavage analysis.

In selected clones (1-2 and 2), total histone levels, including H4, were comparable to those of parental HeLa S3 cells (Fig. EV1B). These H4S47C-expressing clones also displayed normal proliferation rates (Fig. EV1C). RNA-seq analysis confirmed similar gene expression profiles between the clones (Fig. EV1D), with 76.7 and 76.8% of total H4 mRNAs mapped to the H4S47C transgene in clones 1-2 and 2, respectively.

To perform chemical mapping, we synchronized over 90% of HeLa S3 cells in prometaphase using a double thymidine–nocodazole block protocol (Fig. EV1E,F) and established mapping workflows for both interphase and mitotic states (Fig. 1A). Chemical mapping experiments were conducted in clone 2 cells with two replicates for interphase and metaphase (Fig. EV1G), yielding 1.1, 2.0, 0.9, and 1.2 billion uniquely mapped reads for interphase 1, interphase 2, metaphase 1, and metaphase 2, respectively. To evaluate the consistency and robustness of chemical mapping, we first calculated nucleosome center position (NCP) scores, which measure the abundance of nucleosomes with dyads located at each genomic position, as well as the derived center-weighted nucleosome occupancies for both replicates under each condition, following the previously established pipeline (Brogaard et al, 2012). To control for sequencing depth, we randomly subsampled half of the reads from interphase 2 for comparison with interphase 1. Correlation analyses revealed strong agreement between biological duplicates (Fig. EV2A,B). Using the unique nucleosome maps, we quantified center-to-center distances between matched nucleosomes across conditions: interphase 1 vs. interphase 2, metaphase 1 vs. metaphase 2, interphase 1 vs. metaphase 1, and interphase 2 vs. metaphase 2 (Fig. EV2C). We found that 12, 11.8, 8.4, and 9.3% of ~15 million nucleosomes were identically positioned, respectively. Consistent with these results, nucleosome occupancy tracks at individual genomic loci were reproducible across replicates (Fig. EV2D–F). To further improve accuracy and facilitate comparison, we merged the biological replicates to generate consolidated interphase and metaphase chemical maps. The merged maps retained features observed in individual replicates (Fig. EV2D–F).

From the merged data, we defined two types of maps for clone 2: (1) a unique map containing ~15 million well-positioned nucleosomes with a minimum spacing of 120 bp, and (2) a redundant map comprising ~133 million nucleosomes with NCP scores above a defined threshold (see Methods). A Crick-to-Watson strand cleavage peak–peak distance plot (Fig. 1B) revealed conserved cleavage patterns at +2, –5, and –12 nucleotides across the cell cycle, confirming the primary and secondary cleavage sites predominantly occur at position -1 and +6, respectively, relative to the dyad on both strands. Additionally, both maps displayed robust AA/TT/AT/TA dinucleotide periodicity, confirming mapping accuracy (Fig. 1C).

Because PiggyBac transgenes integrate randomly, positional effects could influence the outcome of chemical mapping. To evaluate this possibility, we analyzed an independent clone (1-2), using 1.7 billion and 1.1 billion reads for interphase and mitotic cells, respectively, to construct nucleosome maps. We observed consistent nucleosome occupancy patterns across both clones at representative loci in interphase and mitosis (Fig. 1D–F). For example, Fig. 1D shows a well-positioned nucleosome at a CTCF-binding site during interphase that disappears during mitosis, with adjacent nucleosome arrays adopting an anti-phase pattern. In Fig. 1E, nucleosome positioning changes markedly around the transcription start site (TSS) of the AAR2 gene. In contrast, both interphase and mitosis show similar nucleosome arrangements across the C16orf46 open reading frame (ORF) in Fig. 1F. Correlation analysis confirmed that nucleosome maps obtained from clone 2 and clone 1-2 are in agreement with each

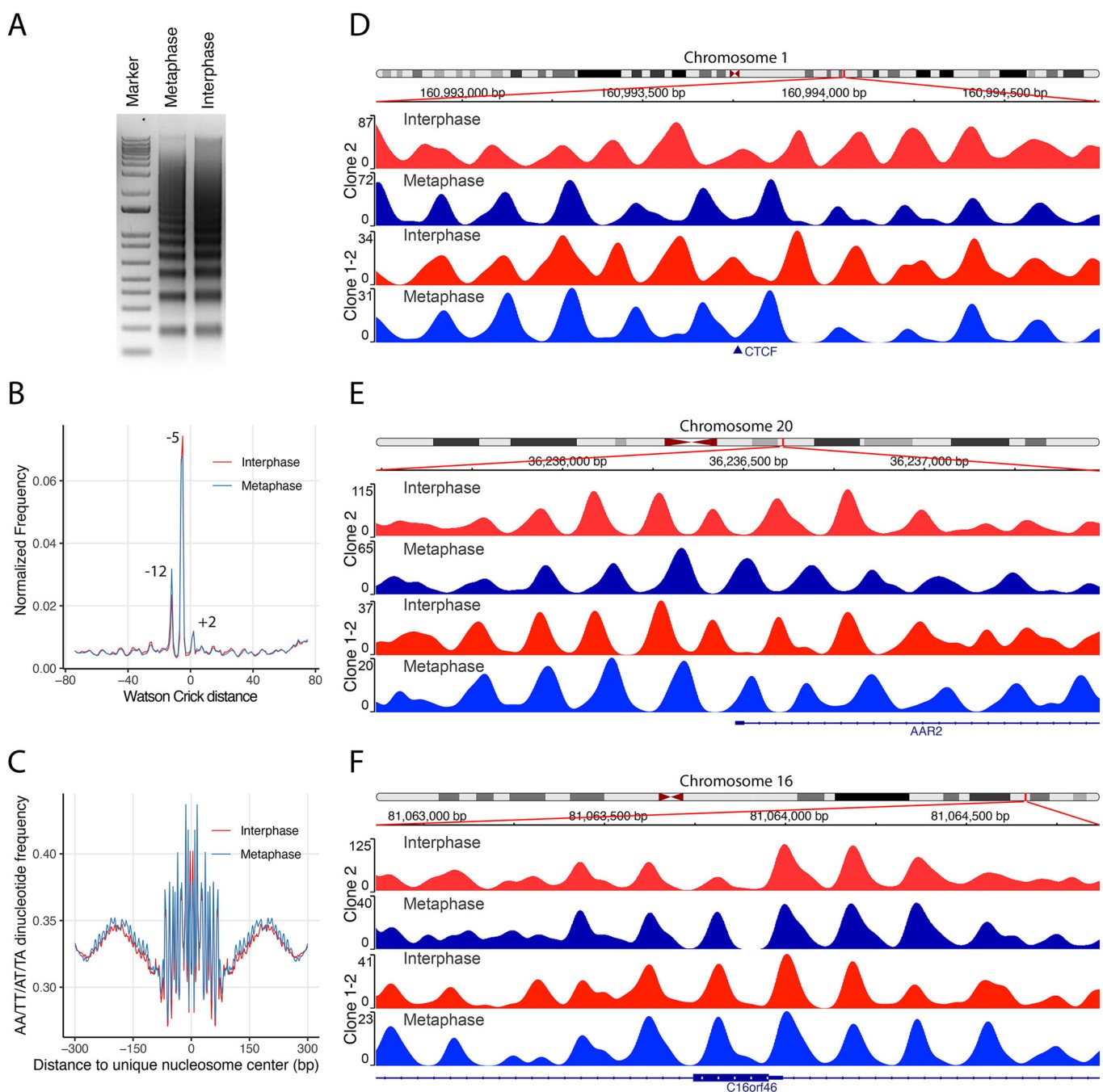

**Figure 1. Chemical nucleosome maps for interphase and mitotic HeLa cells.**

(A) Chemically cleaved nucleosomal DNA fragments from interphase and mitotic HeLa S3 cells visualized on an agarose gel. (B) Crick–Watson cleavage peak-to-peak distance plot showing three dominant distances (−12, −5, and +2 nucleotides), consistent with primary and secondary cleavage sites at −1 and +6, respectively. (C) Frequency of AA/TT/AT/TA dinucleotides within nucleosomes and their flanking regions, based on unique nucleosome maps from interphase and metaphase. (D–F) Representative genomic loci showing nucleosome occupancy scores from both clone 2 and clone 1-2. Distinct interphase–metaphase differences are observed at a CTCF-binding site on Chromosome 1 (D) and the TSS of *AAR2* on Chromosome 20 (E), while similar patterns are found near an exon of *C16orf46* on Chromosome 16 (F). Nucleosome organization is consistent across both clones. Source data are available online for this figure.

other (Appendix Fig. S3), and center-to-center distance analysis revealed that ~17% of unique nucleosome dyads from the two clones are identically positioned in both interphase and metaphase maps (Fig. EV2G), supporting the positional consistency of the two datasets.

Given the consistency between clones, we performed downstream analyses separately for each. We chose to present data from clone 2 in the main text due to its higher sequencing depth, while data from clone 1-2 are included in the appendix (Appendix Figs. S4–S8) for validation.

## Global difference in nucleosome positioning between interphase and metaphase chromosomes

To quantify genome-wide differences in nucleosome landscapes between interphase and metaphase, we calculated Pearson correlation coefficients for nucleosome occupancy scores using a 501-bp sliding window with a 1-bp step size. The empirical cumulative distribution function (ECDF) revealed that over 66% of the genome has a correlation ≥0.5 (Fig. 2A), while a smaller fraction displays substantially altered nucleosome positioning. To localize these differences, we segmented the genome into 3,029,672 consecutive 1-kb bins and computed the Pearson correlation between interphase and metaphase for each bin. These values were then analyzed across 18 chromatin states (E1–E18) defined by ChromHMM based on histone modification patterns (Ernst and Kellis, 2012; Kundaje et al, 2015) (Appendix Table S1). Using a correlation threshold of 0.15, we identified 127,136 low-correlation bins and assessed their enrichment within each chromatin state using odds ratio statistics (Fig. 2B). Low-correlation regions were most enriched (odds ratio >4) in active TSS (E1), flanking TSS (E2–E4), and active enhancer 1 (E9) states (Fig. 2B,C; Appendix Table S1), but reduced (odds ratio <1) in strong transcription (E5), weak transcription (E6), heterochromatin (E13), repressed polycomb (E16), weak repressed polycomb (E17), and quiescent/low (E18). To confirm these patterns independently of cutoff selection, we plotted ECDF curves for each chromatin state (Appendix Fig. S1). ECDF curves for E1–E4 and E9 show the largest deviation above the genome-wide curve in the 0–0.5 correlation range, confirming enrichment for low-correlation regions. In contrast, E13 (heterochromatin) shows the most downward deviation, suggesting the most conserved nucleosome occupancy pattern in the heterochromatin state between interphase and metaphase.

To further investigate global nucleosome rearrangement during mitosis, we analyzed internucleosome linker length distributions in interphase and metaphase (Fig. 2D). Both distributions exhibit a pronounced $10n + 5$ bp periodicity at shorter linker lengths, consistent with prior reports in yeast and mouse (Brogaard et al, 2012; Moyle-Heyrman et al, 2013; Voong et al, 2016). However, notable differences emerged: interphase displayed greater variability in nucleosome spacing, with a pronounced enrichment of short linker lengths (≤30 bp) and reduced frequency in the 30–50 bp range relative to metaphase (Fig. 2D). Linker length profiles across specific chromatin states revealed that regions enriched for low-correlation bins, including transcription start site related states (E1 through E4) and active enhancers (E9 through E11), exhibited even stronger enrichment of short linkers (less than 30 base pairs) during interphase (Fig. 2E). Conversely, states with low enrichment for nucleosome repositioning, such as Strong and Weak Transcription (E5 and E6) and Heterochromatin (E13), showed significantly less enrichment of shorter linker length compared to E1–E4 or E9–E11(Fig. 2D,E, see $p$ values of pairwise comparison in Appendix Table S2). Additional state-specific analyses are provided in Appendix Fig. S2.

Collectively, these findings reveal that while nucleosome organization is broadly preserved between interphase and metaphase, functionally important regulatory regions undergo localized remodeling, marked by denser nucleosome arrays and shorter linker lengths in interphase.

## Nucleosome repositioning at TSS during mitosis

In interphase, actively transcribed genes exhibit a well-positioned nucleosome array downstream of the TSS and an NDR upstream (Baldi et al, 2018; Hughes and Rando, 2014; Lai et al, 2018; Lee et al, 2004). Recent studies suggest NDRs may contain partially unwrapped, fragile nucleosomes induced by nucleosome remodelers or TFs (Brahma and Henikoff, 2020). To identify fragile nucleosomes near the TSS, we compared interphase nucleosome occupancy maps generated using chemical cleavage and micro-coccal nuclease (MNase) digestion in clone 2 HeLa S3 cells (Fig. EV3A). Unlike mESCs(Voong et al, 2016), HeLa cells showed a less prominent -1 nucleosome peak, possibly reflecting reduced numbers of fragile nucleosomes or differing phasing patterns. Focusing on 7721 highly expressed protein-coding genes, we applied K-means clustering to nucleosome occupancy profiles within the [−150, +250] bp window around the TSS, revealing eight distinct clusters (Fig. EV3B). Comparison with MNase data showed that the strong -1 nucleosomes detected by the chemical method in Clusters 2 and 6 were absent in the MNase map, likely due to their fragility (Fig. EV3B).

To investigate nucleosome dynamics near active transcription units during mitosis, we first plotted the average local correlation within ±1000 bp of transcription start sites (TSS) and transcription termination sites (TTS), dividing 19,166 protein-coding genes into five expression-based groups: no expression, bottom 25%, 25–50% (low–moderate), 50–75% (moderate–high), and top 25% (high) based on interphase RNA levels (Fig. 3A). The local correlation plots revealed a V-shaped dip centered at the TSS, where higher gene expression levels were generally associated with lower nucleosome correlation between interphase and metaphase, except that the top 25–50% and top 25% expression groups exhibited very similar correlation patterns, but in reverse order. In contrast, the TTS region exhibited an opposite and more consistent trend: higher expression levels corresponded to higher correlation values. We next analyzed nucleosome occupancy patterns around the TSS and TTS in both interphase and metaphase, stratified by the same five expression groups (Fig. 3B,C). Around the TSS, nucleosome occupancy was generally higher in interphase than in metaphase (Fig. 3B). In interphase, genes with higher expression levels displayed stronger nucleosome phasing downstream of the TSS, a pattern that was absent in metaphase (Fig. 3C). At the TTS, both interphase and metaphase maps exhibited similar phasing patterns, with nucleosome occupancy positively correlating with gene expression levels. To further explore these differences, we compared the eight TSS-centered nucleosome clusters identified in interphase (Fig. EV3B) with their corresponding average occupancy patterns in metaphase (Fig. 3D). Unlike the strongly phased arrays observed in interphase, metaphase nucleosomes lacked such organization near the TSS, indicating significant rearrangement during mitosis.

We then calculated linker length distributions within ±1000 bp of the annotated TSSs of 15,887 expressed protein-coding genes (Fig. 3E). This analysis confirmed a significantly greater enrichment of shorter linker lengths (≤30 bp) around the TSS in interphase compared to metaphase than the genome average (Fig. 2D; Appendix Table S3). To visualize how these reduced linker lengths relate to nucleosome positioning, we examined individual occupancy profiles in representative promoters. As shown for loci on chromosomes 1, 19, and 20, interphase TSS regions often exhibited denser nucleosome arrays, consistent with shorter

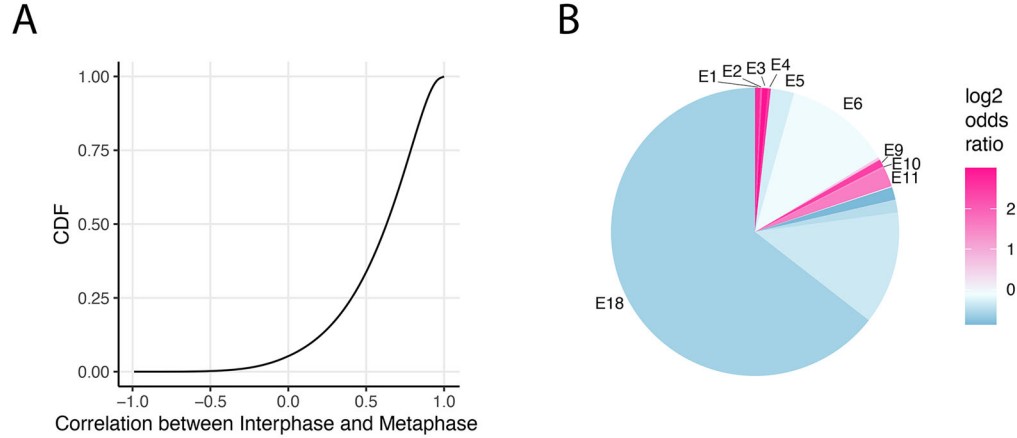

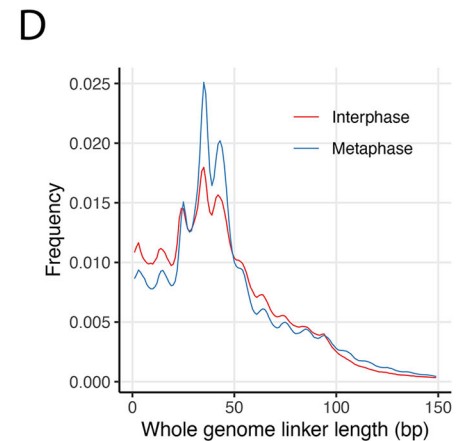

C

| | | Correlation ≤ 0.15 | Correlation > 0.15 | Abundance Ratio | Odds Ratio |
|---|---|---|---|---|---|
| Chromatin Stats E1 | ROI | 3783 | 12238 | 30.91% | 6.55 |
| | BG | 123353 | 2612263 | 4.72% | |
| Chromatin Stats E2 | ROI | 1018 | 4802 | 21.20% | 4.40 |
| | BG | 126118 | 2619699 | 4.81% | |
| Chromatin Stats E3 | ROI | 4794 | 12676 | 37.82% | 8.07 |
| | BG | 122342 | 2611825 | 4.68% | |
| Chromatin Stats E4 | ROI | 1944 | 7262 | 26.77% | 5.60 |
| | BG | 125192 | 2617239 | 4.78% | |
| Chromatin Stats E9 | ROI | 4359 | 17429 | 25.01% | 5.31 |
| | BG | 122777 | 2607072 | 4.71% | |

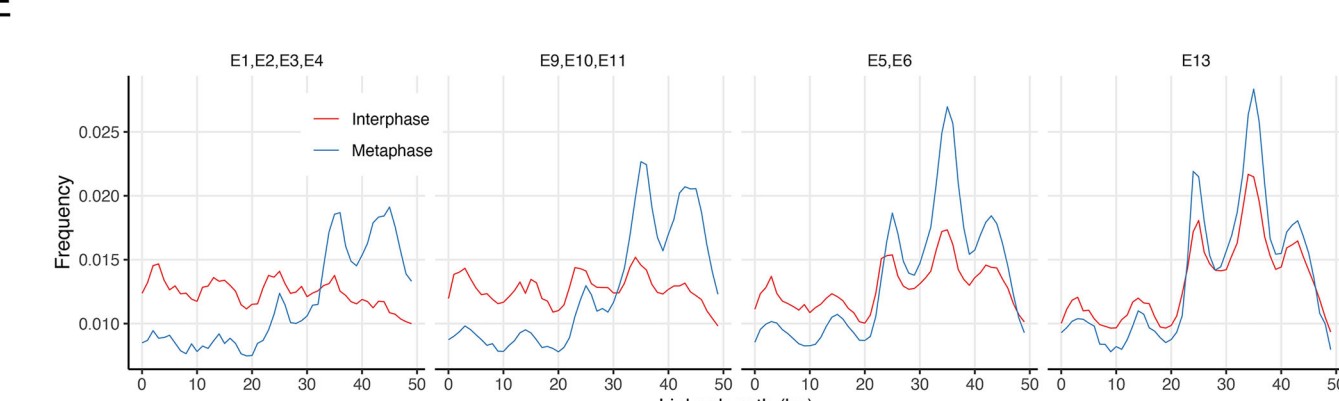

**Figure 2.  Global analysis of nucleosome reorganization between interphase and metaphase chromosomes.**

(A) Empirical cumulative distribution function (ECDF) plot showing genome-wide local correlation in nucleosome occupancy scores between interphase and metaphase. Correlations were calculated using a 501-bp sliding window with a 1-bp step size. (B) Pie chart showing the proportion of the genome covered by ChromHMM-defined chromatin states. Color intensity reflects the $\log_2$ odds ratio of a bin being classified as low correlation (bin-correlation ≤0.15) within each state relative to the rest of the genome. The genome was divided into 3,029,672 non-overlapping 1000-bp bins. Of these, 2,751,637 bins had valid correlation values, and 127,136 were classified as low correlation. (C) Table summarizing the number of low- and high-correlation bins, odds, and $\log_2$ odds ratios for chromatin states E1, E2, E3, E4, and E9. ROIs (regions of interest) correspond to bins annotated with the indicated chromatin state; BG (background) includes all other bins in the genome. (D) Linker length distributions derived from unique nucleosome maps, showing increased variability and a shift toward shorter linkers in interphase compared to metaphase. (E) Linker length profiles across selected chromatin states, comparing interphase and metaphase. Shown are transcription start site-flanking or upstream/downstream regions (E1–E4), active and weak enhancers (E9–E11), strong and weak transcription (E5, E6), and heterochromatin (E13). Source data are available online for this figure.

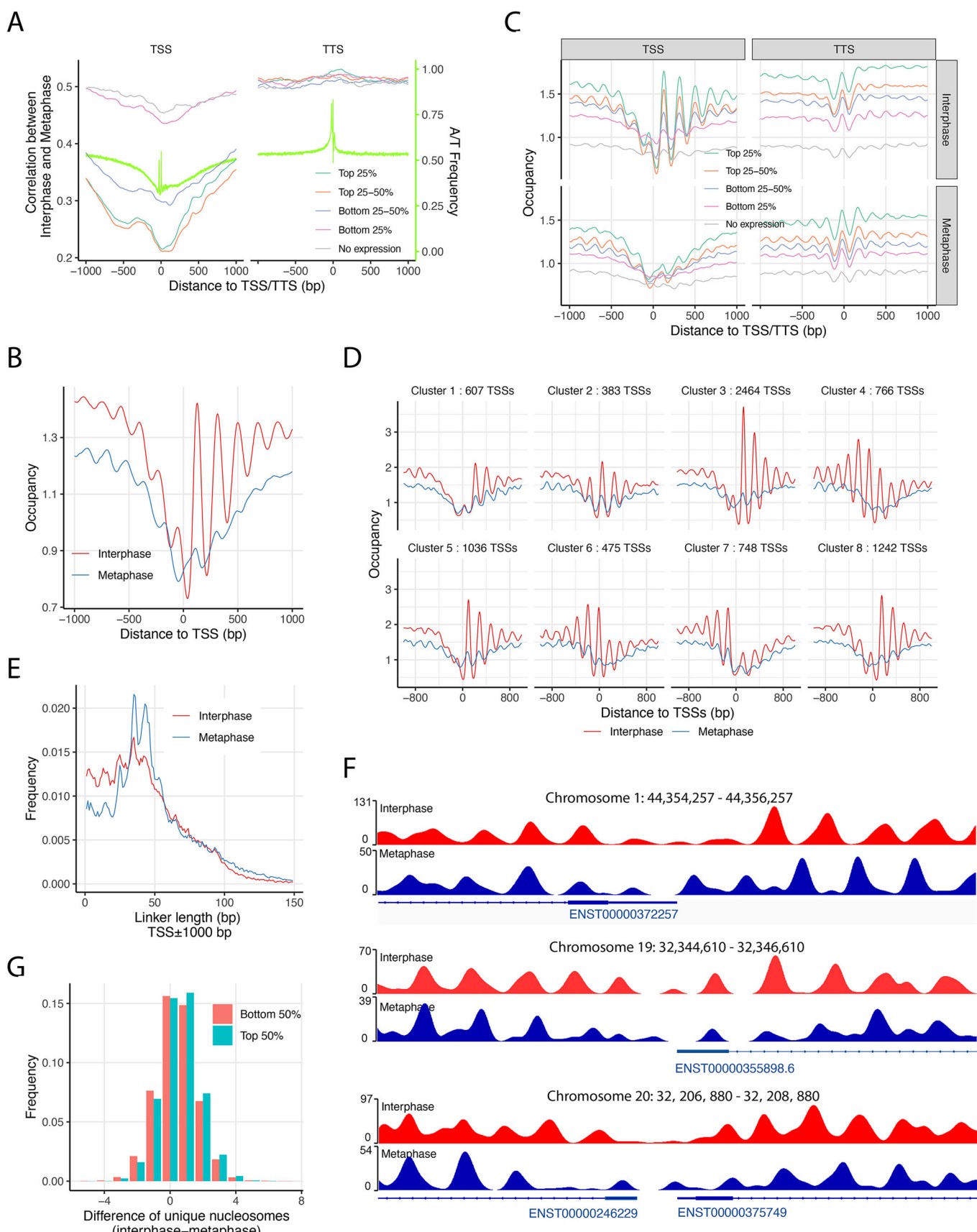

**Figure 3. Nucleosome array repositioning around active promoters during mitosis.**

**(A)** Local correlation between interphase and metaphase nucleosome occupancy scores at transcription start sites (TSSs) and transcription termination sites (TTSs), averaged and grouped by gene expression quartiles from 19,166 protein-coding genes. **(B)** Average nucleosome occupancy profiles at the TSSs of 15,887 expressed protein-coding genes in interphase and metaphase. **(C)** Nucleosome occupancy scores at TSSs and TTSs from 19,166 genes, categorized into quartiles according to gene expression levels. **(D)** A total of 7944 TSSs from the top 50% expressed genes were grouped into eight clusters using K-means clustering, based on interphase occupancy scores in the [−150, +250] bp window around the TSS. TSSs ($n = 464$) without valid occupancy scores in either interphase or metaphase were excluded. **(E)** Linker length distributions near TSSs, showing an enrichment of shorter linkers in promoter regions during interphase. **(F)** Representative examples of nucleosome repositioning near the TSSs of *ERI3* (ENST00000372257, top), *ZNF507* (ENST00000355898, middle), *PLAGL2* (ENST00000246229) and *POFUT1* (ENST00000375749, bottom), demonstrating nucleosome shifts and additions in interphase compared to metaphase. **(G)** Histogram showing the difference in unique nucleosome counts within ±1000 bp of each TSS between interphase and metaphase, for genes in the top and bottom 50% of expression levels. Source data are available online for this figure.

internucleosomal spacing (Fig. 3F). To validate this trend, we quantified the difference in nucleosome counts within ±1000 bp of the TSS between interphase and metaphase for genes in the top and bottom halves of expression (Fig. 3G). On average, there were 0.570 and 0.454 more unique nucleosomes in interphase than metaphase for the top and bottom expression halves, respectively, suggesting that nucleosome density at the TSS is positively associated with transcription activity. This is consistent with a previous report showing that internucleosome spacing decreases with increased transcription (Valouev et al, 2011).

In summary, our chemical mapping data suggest that during interphase, transcription imposes a robust barrier effect at the TSS, generating synchronized nucleosome phasing downstream and, to a lesser extent, upstream. In mitosis, this barrier effect is diminished, leading to increased internucleosomal spacing and repositioning, resulting in less dense and desynchronized nucleosome arrays surrounding the TSS.

## Differential nucleosome positioning at enhancers in interphase and metaphase chromosomes

We next examined nucleosome positioning at enhancers for two key reasons. First, enhancer-promoter interactions are crucial for transcriptional regulation (Thurman et al, 2012), yet this long-range communication is disrupted during mitosis (Ito and Zaret, 2022). However, how mitosis affects local nucleosome organization at these sites remains unclear. Second, enhancers are often associated with NDRs (Barozzi et al, 2014; Klemm et al, 2019; Lai et al, 2018; Lee et al, 2004). Previous MNase-based studies lacked sufficient resolution to accurately identify fragile nucleosomes at enhancers (Brahma and Henikoff, 2020). In this study, we defined active enhancers in HeLa S3 cells by overlapping enhancer regions classified as E9 (active enhancer 1) and E10 (active enhancer 2) chromatin states with a curated set of human enhancer annotations derived from diverse cell types and tissues (Leung et al, 2015). This approach yielded a refined set of 9173 active enhancer sites.

To characterize nucleosome organization at these enhancers, we plotted the average local correlation and nucleosome occupancy around enhancer centers (Fig. 4A,B). On average, we observed only a modest dip in local correlation and slightly higher nucleosome occupancy in interphase compared to metaphase. Given the substantial enrichment of low-correlation signals at enhancer regions, we hypothesized that local correlation patterns may vary considerably between individual sites. To investigate this, we applied K-means clustering to the local correlation profiles within ±400 bp of each enhancer center and visualized the cluster centroids and corresponding heatmap (Fig. 4C,D). Six clusters

(Clusters 1–6), comprising ~7792 sites, exhibited pronounced drops in local correlation (from ~0.6 to 0.1–0.2), with the position of the trough varying across the ±400 bp window. In contrast, the remaining two clusters displayed peaks in correlation centered at enhancer sites, with only minor decreases in the flanking regions.

Two main factors may contribute to the observed variability in local correlation patterns at enhancer sites. First, inaccuracies in the curated locations of enhancers could lead to misalignment of nucleosome features. Second, enhancers can be defined through multiple, cell-type-specific mechanisms, each with differing requirements for precise nucleosome positioning. These sources of heterogeneity likely obscure well-defined nucleosome phasing when enhancer sites are analyzed in aggregate. To illustrate how such variability is captured by local correlation analysis, we selected two representative enhancer sites from clusters 2 and 5. The first example, located on Chromosome 2, displays low-correlation upstream of the enhancer site, which is indicative of differential nucleosome positioning between interphase and metaphase, while showing strong and consistent phasing downstream (Fig. 4E). In contrast, the second example from Chromosome 6 exhibits the opposite pattern, with conserved phasing upstream and variable positioning downstream of the enhancer center (Fig. 4F).

For the active enhancer set, we analyzed the linker length distribution using unique nucleosomes located within ±1000 bp of each site. This analysis revealed even more enriched shorter linker lengths (≤30 bp) during interphase than the genome average (Fig. 4G; Appendix Table S3), consistent with patterns observed in Chromatin E9–E11 states (Fig. 2E). We next quantified changes in nucleosome occupancy by calculating the difference in unique nucleosome counts between interphase and metaphase within the same ±1000 bp window (Fig. 4H). This revealed that during interphase, over 43% of enhancer regions gained 1–2 unique nucleosomes, ~32% remained unchanged, and ~21% exhibited a decrease in nucleosome count. On average, 0.376 more nucleosomes were detected per enhancer in interphase relative to metaphase across the 2-kb window (Fig. 4H). Collectively, these results indicate that in response to mitotic chromosome condensation and transcriptional silencing, distal enhancers undergo extensive nucleosome repositioning—characterized by lengthened linker regions and reduced nucleosome density—paralleling the chromatin reorganization observed at promoter regions.

## Differential nucleosome phasing patterns at CTCF sites during mitosis

During prometaphase, CTCF is evicted from chromatin (Oomen et al, 2019; Zhang et al, 2019). Previous studies have shown that

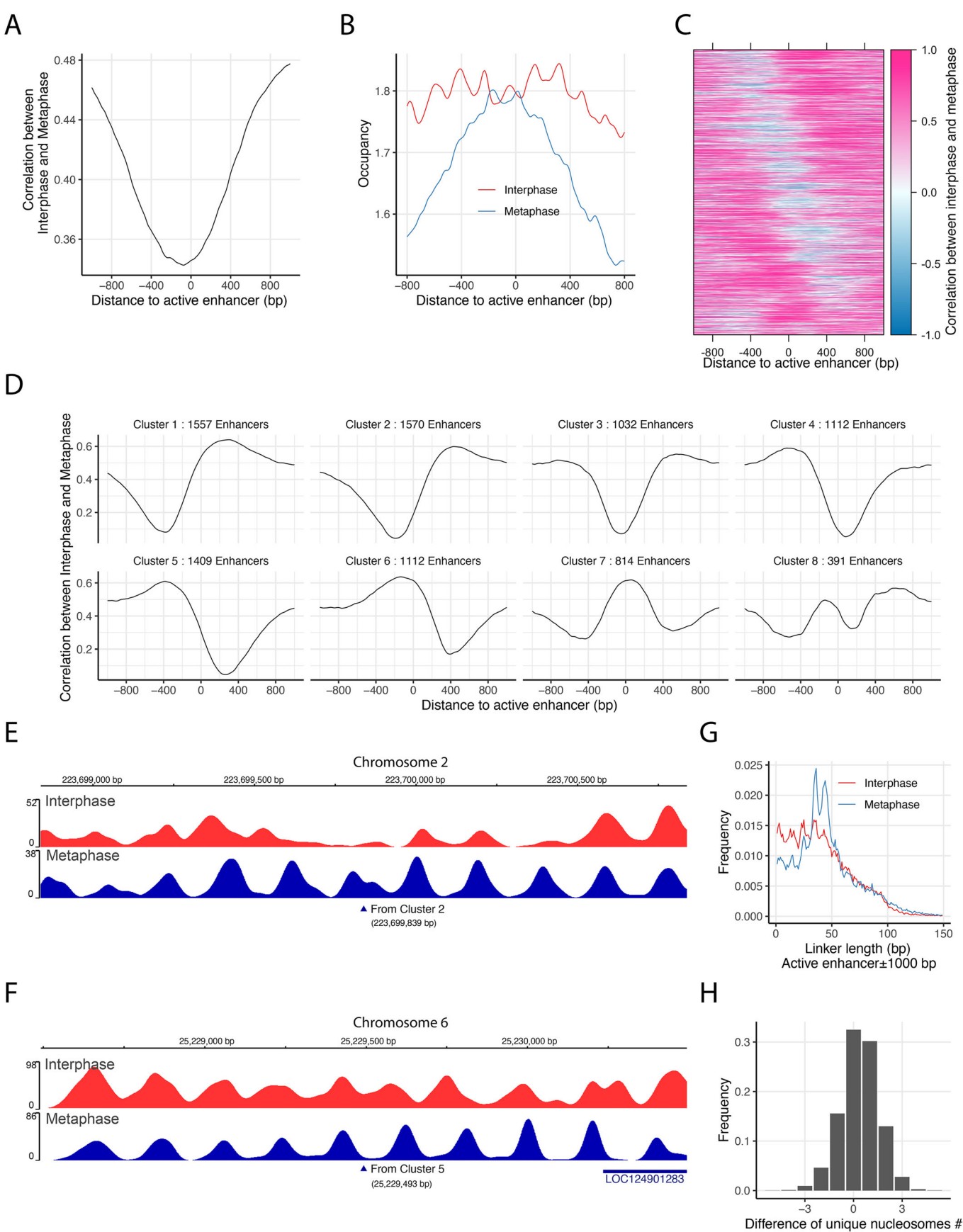

**Figure 4.  Nucleosome organization at active enhancers during mitosis.**

(A) Average local correlation between nucleosome occupancy scores from interphase and metaphase maps, aligned at 9173 active enhancer sites. (B) Comparison of average nucleosome occupancy profiles surrounding active enhancers in interphase and metaphase. (C, D) K-means clustering was performed on local correlation values within ±400 bp of active enhancer sites. (C) Heatmap of the clustered local correlation profiles. (D) Centroid plots representing the average correlation profiles of the eight resulting clusters. (E, F) Representative examples of active enhancers exhibiting differential nucleosome organization between interphase and metaphase: one on Chromosome 2 (Cluster 2) (E), and another on Chromosome 6 (Cluster 5) (F). (G) Linker length distributions near active enhancers in interphase versus metaphase. (H) Histogram showing the difference in unique nucleosome counts within ±1000 bp of each active enhancer between interphase and metaphase. On average, interphase exhibits 0.376 more unique nucleosomes per enhancer than metaphase. Source data are available online for this figure.

nucleosomes flanking CTCF sites reposition towards the CTCF-binding site upon CTCF eviction on mitotic chromosomes (Oomen et al, 2019; Owens et al, 2019). In these studies, CTCF sites were located within the NDRs. To further substantiate these observations, we analyzed 74,550 CTCF-binding sites identified from a CTCF ChIP-seq dataset in interphase HeLa S3 cells (Ram et al, 2011). Enrichment analysis revealed a significant association between CTCF sites and low interphase–metaphase correlation regions, with an odds ratio of 5.59. We divided the CTCF sites into four quartiles based on their ChIP-seq fold enrichment scores and observed a consistent V-shaped pattern in average correlation, with the lowest correlation at the center of CTCF sites (Fig. 5A). Within ±1000 bp of CTCF sites, linker lengths ≤30 bp were more enriched compared to the genome average (Appendix Table S3), as shown in Fig. 5B. We then assessed average nucleosome occupancy around CTCF sites in interphase and metaphase (Fig. 5C), grouped them by ChIP-seq quartiles (Fig. 5D), and displayed nucleosome occupancy at individual CTCF sites in a heatmap sorted by descending ChIP-seq enrichment (Fig. 5E). Together, these analyses show that CTCF sites in interphase HeLa cells are occupied by well-positioned nucleosomes, consistent with previous findings in mouse ESCs (Voong et al, 2016). In contrast, this pattern is lost during metaphase. Nevertheless, in both cell cycle states, CTCF sites act as a bidirectional barrier that organizes phased nucleosome arrays in the surrounding regions. However, the absence of a nucleosome at the CTCF site during metaphase results in anti-phase nucleosome positioning at the CTCF center between the two states. This phase shift gradually dissipates, with nucleosome phasing becoming synchronized approximately ±700 bp away. Importantly, the strength of the barrier effect correlates with CTCF ChIP-seq signal intensity (Fig. 5D,E), leading to denser nucleosome arrays and confirming the shorter nucleosome spacing observed in interphase (Fig. 5B).

## Nucleosome positioning is preserved at exon/intron boundaries during mitosis

ChromHMM analysis revealed that the strong transcription (E5) and weak transcription (E6) chromatin states exhibit low odds ratio scores (Appendix Table S1), indicating consistent behavior throughout the cell cycle. These states are marked by H3K36me signals and are typically associated with actively transcribed exons within gene bodies (Andersson et al, 2009; Kolasinska-Zwierz et al, 2009). To examine nucleosome occupancy at exonic regions, we analyzed 164,916 non-terminal exons from 19,166 human protein-coding genes. In interphase, chemical maps revealed clear enrichment of nucleosomes at exon boundaries, with nucleosome occupancy positively correlating with gene expression levels

(Fig. 6A, top). In contrast, MNase maps lacked the resolution to resolve phased nucleosomes and instead showed broader, less defined peaks centered over exons (Fig. 6B, top) (Schwartz et al, 2009; Tilgner et al, 2009).

Surprisingly, despite the global reduction in transcription during mitosis, nucleosome positioning patterns around exons are largely preserved (Fig. 6A,B, bottom). To further investigate the relationship between transcriptional activity and nucleosome positioning at exon–intron boundaries, we analyzed NET-seq (native elongating transcript sequencing) data from a previously published study (Mayer et al, 2015). We plotted NET-seq read densities alongside nucleosome center positioning (NCP) scores at exon–intron and intron–exon junctions, respectively (Fig. 6C,D). Consistent with prior reports (Mayer et al, 2015), NET-seq data showed an increasing accumulation of transcribing RNA polymerase II (RNAPII) at both junctions. Notably, NCP score peaks coincided with NET-seq signal peaks, indicating enrichment of nucleosome dyads at exon boundaries (Fig. 6C). This pattern was observed across expression-level stratified gene groups (Fig. 6D), suggesting a consistent association. These observations, together with our previous findings in mESCs (Voong et al, 2016), support a model in which nucleosomes act as physical barriers that modulate RNAPII progression at exon–intron junctions, contributing to exon definition in human cells. Importantly, the chemical maps further demonstrate that these characteristic nucleosome positioning patterns persist in mitotic cells. Given that transcription and splicing machinery are largely absent from metaphase chromosomes, the persistence of these signals implies that they are intrinsically encoded by DNA sequence. Thus, the observed RNAPII pausing at exon–intron boundaries in interphase (Kwak et al, 2013; Mayer et al, 2015) likely reflects, at least in part, a structural contribution from nucleosomes, rather than being solely a consequence of RNAPII behavior.

## The role of intrinsic DNA bendability in dynamic nucleosome positioning

Using the human chemical nucleosome maps, we have characterized dynamic changes in nucleosome positioning between interphase and metaphase chromosomes. A consistent trend emerges during the mitotic-to-interphase transition: at regulatory elements such as promoters, enhancers, and insulators, interphase nucleosome arrays exhibit substantial reorganization. This is marked by shortened linker lengths, the incorporation of fragile nucleosomes, and increased nucleosome density. In contrast, nucleosome positions across much of the genome—particularly within coding regions such as exons—remain relatively stable between the two cell cycle stages. This contrast offers a unique opportunity to investigate

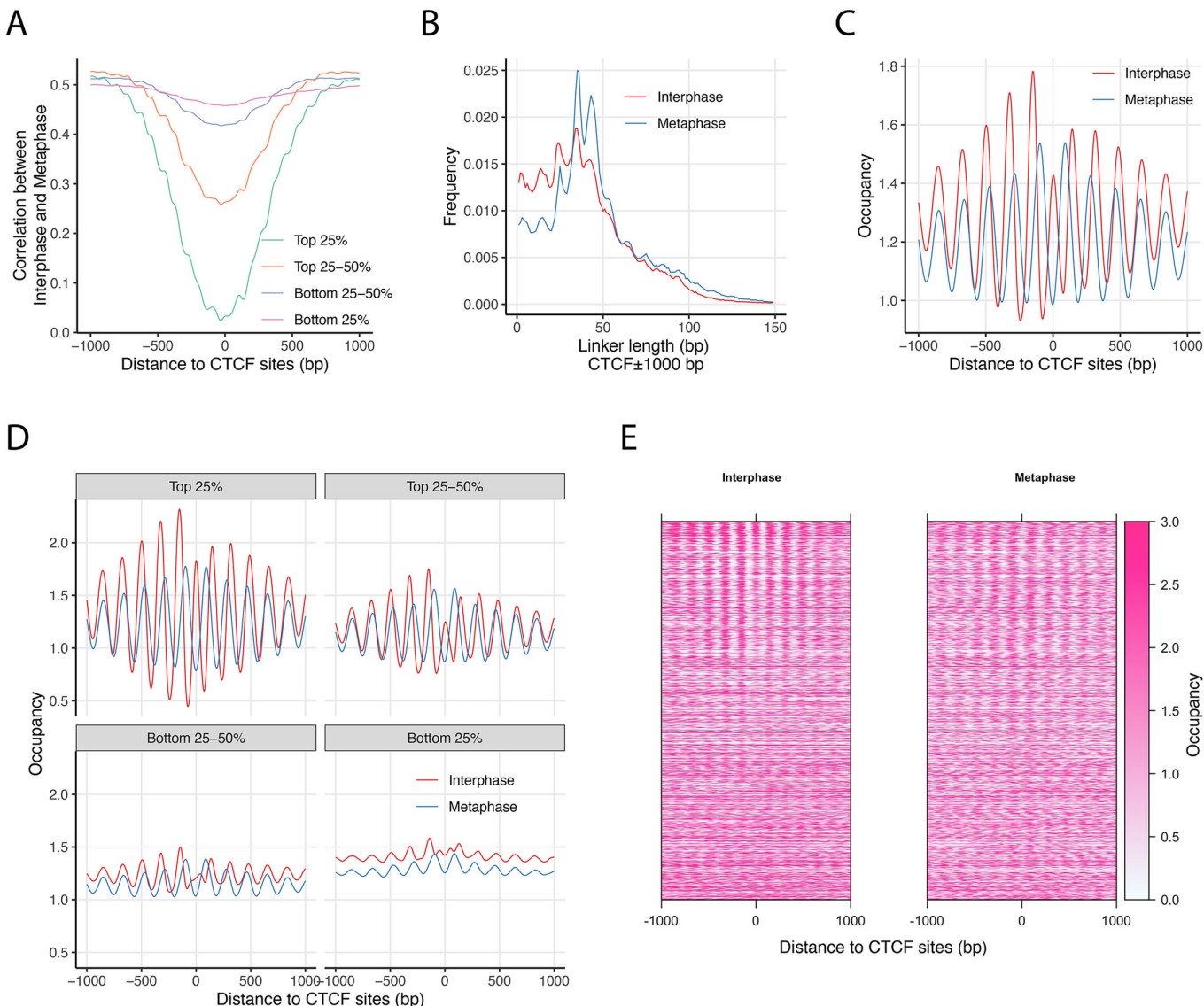

**Figure 5. Nucleosome positioning at CTCF-binding sites during mitosis.**

(A) Local correlation analysis of nucleosome occupancy scores at CTCF-binding sites based on interphase and mitotic chemical maps. Data were divided into quartiles according to CTCF ChIP-seq signal intensity (GEO: GSM733785). (B) Comparison of linker length distributions near CTCF-binding sites in interphase and metaphase genomes. (C) Nucleosome occupancy scores at CTCF-binding sites from interphase and metaphase chemical maps. (D) Extended analysis of panel (C), with nucleosome occupancy scores further divided into quartiles based on CTCF ChIP-seq signals. (E) Heatmaps of nucleosome occupancy scores from interphase and mitotic maps, sorted in descending order of CTCF ChIP-seq signal at binding sites. Source data are available online for this figure.

the contribution of intrinsic DNA bendability to nucleosome dynamics during mitosis.

Using DNAcycP2 (Kendall et al, 2025), we computed the DNA cyclizability (C-score) across the entire human genome. Each C-score reflects the predicted DNA bendability of a 50-bp window centered at a given genomic location. We aggregated the C-scores within ±1 kb of unique nucleosome dyad positions from both interphase and metaphase maps (Fig. 7A). The average C-score profile displays a phasing pattern consistent with nucleosome organization, suggesting a general preference for highly bendable DNA in nucleosome-bound regions. Within nucleosomes, cyclizability is not evenly distributed. A broad peak in C-score is

observed around the dyad (±20 bp) in both interphase and metaphase, consistent with structural models of tight DNA–histone interactions at the dyad. Interestingly, during interphase, two additional broad peaks in DNA cyclizability emerge ~42–73 bp from the dyad, in the nucleosome shoulder regions. These shoulder peaks are significantly reduced in metaphase (Fig. 7B), suggesting cell cycle-dependent alterations of DNA–histone contacts.

To investigate whether specific regulatory elements exhibit distinct DNA bendability preferences in relation to nucleosome positioning, we examined the C-score profiles around unique nucleosome dyads located within ±500 bp of transcription start sites (TSSs), enhancers, and CTCF-binding sites (Fig. 7C–E). As

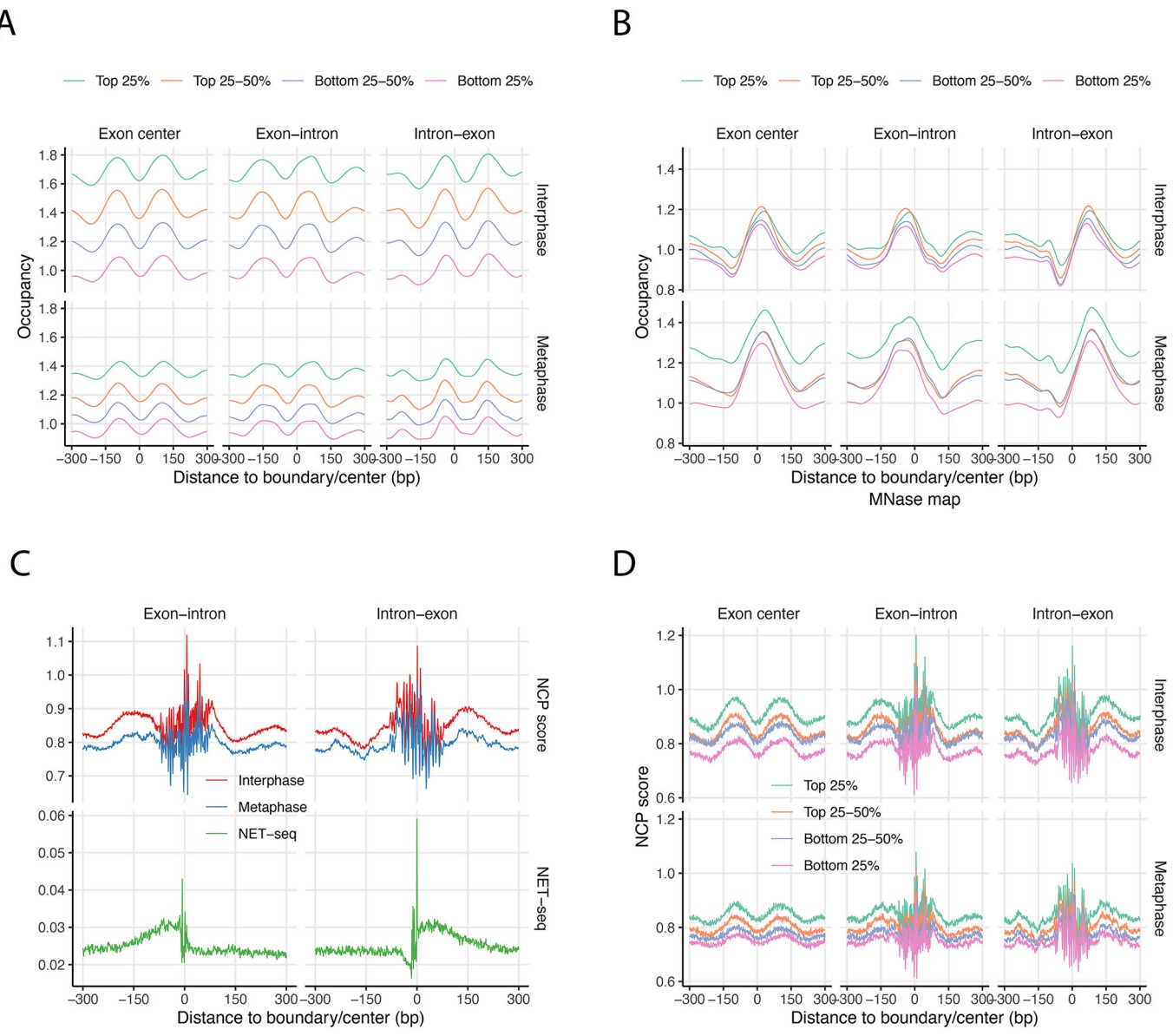

**Figure 6. Nucleosome positioning at exon-intron and intron-exon junctions.**

(A) Nucleosome occupancy scores from chemical maps of interphase (upper panels) and metaphase (lower panels), averaged at exon centers, exon-intron junctions, and intron-exon junctions. A total of 164,916 internal exons from 19,166 protein-coding genes were analyzed and grouped into quartiles based on gene expression levels (FPKM). About 22,916 exons from non-expressed genes were excluded. (B) Same as panel (A) but using MNase-derived nucleosome maps. (C) NCP scores from interphase (red) and metaphase (blue) chemical maps at exon-intron and intron-exon junctions (upper panels), alongside NET-seq data averaged at the same positions (lower panels). (D) Chemical NCP scores from interphase (upper panels) and metaphase (lower panels) averaged at exon centers and junctions, stratified into expression-level quartiles based on gene FPKM values. Source data are available online for this figure.

CTCF-binding sites are known to have intrinsically high cyclizability (Basu et al, 2022; Li et al, 2022) and are occupied by well-positioned nucleosomes in interphase but not metaphase (Fig. 5C), we further analyzed nucleosomes located at least ±60 bp from the CTCF motif center to assess how local DNA mechanics may influence repositioning (Fig. 7F). Several key findings emerged. First, TSSs, enhancers, and CTCF sites all exhibit elevated average cyclizability compared to the genome-wide background (Fig. 7B). Second, during metaphase, nucleosomes in these regulatory regions show the highest cyclizability near the dyad. In contrast, during

interphase, cyclizability peaks are more pronounced at the shoulder regions that flank the dyad, particularly at transcription start sites, enhancers, and in nucleosomes flanking CTCF sites. (Fig. 7C,D,F). This interphase-specific pattern contrasts with the genome-wide trend and suggests that dynamic chromatin states may leverage DNA mechanics to promote nucleosome repositioning at regulatory elements.

To ensure the generalizability of these conclusions, we repeated the entire analytical pipeline using chemical maps derived from an independent HeLa S3 cell clone (clone 1-2) and observed highly

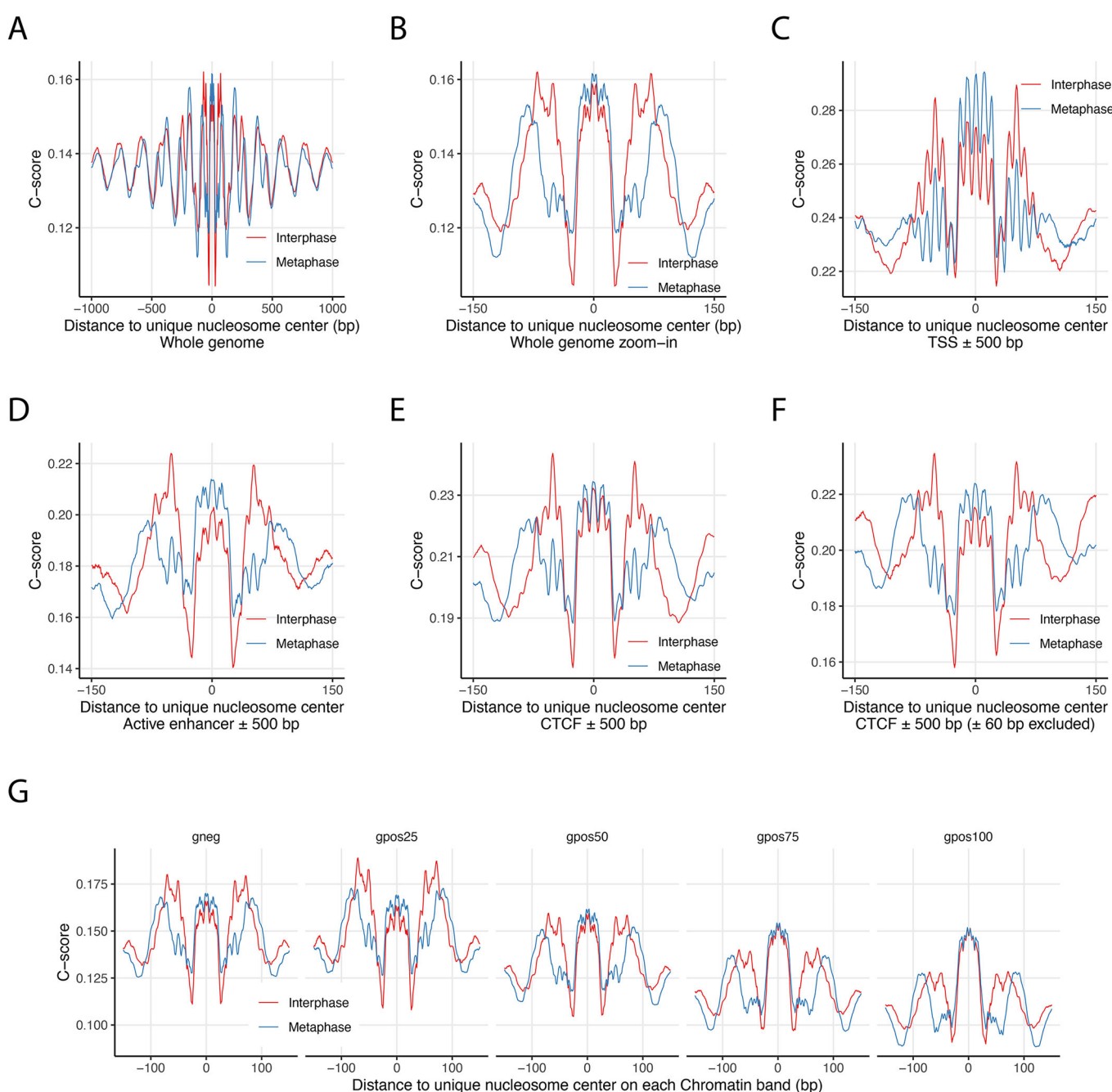

**Figure 7. DNA cyclizability and nucleosome repositioning during mitosis.**

(A) Genome-wide average DNA cyclizability scores (C-scores) for unique nucleosomes and their flanking regions in interphase and metaphase, derived from chemical mapping data. (B) Close-up comparison of average C-scores within the nucleosome region between interphase and metaphase, corresponding to panel (A). (C, D) Nucleosomal C-scores in interphase and metaphase within ±500 bp of TSSs (C) and active enhancers (D). (E) Comparison of C-scores for nucleosomes located within ±500 bp of CTCF-binding sites. (F) Same as panel (E) but restricted to nucleosomes positioned at least 60 bp away from the CTCF motif center. (G) Average nucleosomal C-scores plotted across chromosome regions grouped by Giemsa staining categories. Source data are available online for this figure.

similar results (Appendix Fig. S8). It is important to note that the observed features represent population-averaged trends and do not imply that individual nucleosomes simultaneously exhibit elevated cyclizability at both the dyad and shoulder regions. To explore this variability, we applied K-means clustering (k = 14) to the C-score profiles of individual nucleosomes located near transcription start sites (TSSs). Most resulting clusters displayed one or two dominant peaks in cyclizability across the nucleosomal region, with enrichment either at the dyad or in the shoulder regions (Fig. EV4). These results highlight the heterogeneous nature of local

DNA–nucleosome interactions, emphasizing the nuanced contribution of intrinsic DNA mechanics to nucleosome positioning.

Our results support the interpretation that DNA's intrinsic bendability plays a more significant role in determining nucleosome dyad positioning in the mitotic genome, when transcription and other gene regulatory activities are largely inactive. In this context, DNA segments with higher cyclizability tend to align with the H3–H4 tetramer at the nucleosome dyad, promoting stable, energetically favorable interactions. In contrast, during interphase, nucleosomes often adopt alternative dyad positions that facilitate access to cis-regulatory elements involved in transcription. In such cases, highly cyclizable DNA sequences are more frequently positioned asymmetrically on one side of the nucleosome, away from the dyad.

To assess whether this pattern extends genome-wide across chromatin types, we used Giemsa-stained bands from human metaphase chromosomes as a proxy to distinguish euchromatin and heterochromatin regions (Cheung et al, 2001; Furey and Haussler, 2003). In Giemsa staining, G-negative (gneg) and G-light (gpos25) bands correspond to gene-rich euchromatin, while darker bands (gpos50, gpos75, and gpos100) represent heterochromatin. Analysis of DNA cyclizability preferences in nucleosomes across these G-bands revealed a monotonic decrease in average C-score and in the relative height of shoulder cyclizability peaks compared to dyad peaks, with increasing G-band intensity (Fig. 7G; Appendix Fig. S8G). Notably, strong cyclizability signals are concentrated near the dyad in heterochromatin. In contrast, in euchromatin, especially during interphase, peaks in the shoulder C-score become more prominent and can even exceed those at the dyad (Fig. 7G). These findings suggest that preferential positioning of bendable DNA at one shoulder of the nucleosome is more common in euchromatin, likely reflecting the dynamic chromatin architecture required for active transcription.

Across all individual human chromosomes, we observed a consistent trend in which interphase nucleosomal DNA exhibits higher cyclizability scores at the shoulders compared to metaphase, although the average C-scores vary by chromosome (Appendix Fig. S9). Notably, chromosomes 17, 19, and 22 display particularly elevated shoulder C-scores relative to the dyad, even during metaphase. These chromosomes also have the highest G/C content and gene densities among all human chromosomes (Gilbert et al, 2004). While prior studies indicate that DNA cyclizability is not directly correlated with G/C content (Basu et al, 2022; Basu et al, 2021), gene-rich regions may influence local DNA mechanics in other ways.

Different G-bands vary in gene content, with the trend: gpos25 > gneg > gpos50 > gpos75 > gpos100 (Serna-Pujol et al, 2021). To explore the relationship between gene density and nucleosomal DNA bendability, we analyzed C-score patterns across G-bands on individual chromosomes (Fig. EV5A). Most G-band regions resemble the genome-wide trend or follow the pattern of chromosome 1, which is representative of the majority. However, several regions on specific chromosomes exhibit the same distinctive C-score pattern seen on chromosomes 17, 19, and 22. These include: gpos25 on chromosomes 11, 16, 17, and 19; gneg on chromosomes 16, 17, 19, and 22; and gpos50 on chromosome 17 (Fig. EV5A). Plotting gene densities for individual G-band regions revealed that these regions consistently contain the highest number

of genes per megabase (highlighted as red dots in Fig. EV5B). Importantly, their distinct C-score patterns across the nucleosome region cannot be explained by the average C-scores within that region (Fig. EV5C). Together, these findings suggest that the preferential localization of highly bendable DNA at nucleosome shoulders correlates with local gene density. In line with our broader observations that DNA cyclizability differentially influences nucleosome positioning in genomic regions involved in gene regulation during mitosis (Fig. 7), the local abundance of regulatory elements may play a key role in shaping nucleosome organization based on DNA mechanical properties.

## Discussion

Nucleosome positioning undergoes dynamic changes during various biological processes. Despite prior studies, the exact details of nucleosome reorganization during mitosis remain unclear. Recently developed chemical cleavage-based strategies offer the resolution needed to address this challenge (Brogaard et al, 2012; Voong et al, 2016). In this study, we present the first single-base pair resolution nucleosome maps of the human genome in interphase and metaphase. Systematic analysis of these datasets reveals a unifying theme of nucleosome organization during mitosis. Compared to metaphase chromosomes, interphase chromatin exhibits substantial nucleosome repositioning—characterized by shortened linker lengths, the appearance of fragile nucleosomes, and increased nucleosome density at regulatory regions—to accommodate transcriptional activity (Valouev et al, 2011). These findings provide new perspectives on the organizing principles of the human genome at the local nucleosome level.

Compared to the widely used MNase methods, chemical mapping offers significant advantages in accurately positioning fragile nucleosomes, particularly within regions previously designated as nucleosome-depleted regions (NDRs) (Brahma and Henikoff, 2020). Similar to mESC chemical data (Voong et al, 2016), the human interphase chemical map in HeLa cells identifies fragile nucleosomes at promoters, enhancers, and insulators—regions long presumed to be nucleosome-free. This observation, supported by earlier reports (Kubik et al, 2015; Ramachandran et al, 2017; Voong et al, 2016; Xi et al, 2011), reinforces the conserved nature of fragile nucleosomes at key regulatory elements across species (Brahma and Henikoff, 2020). While chemical mapping does not reveal the origin of fragile nucleosomes, genomic data suggest that NDRs are commonly co-occupied by nucleosome remodelers and TFs (Ahmad et al, 2024; Brahma and Henikoff, 2020; Bulyk et al, 2023). Recent structural studies demonstrate that DNA unwrapping or distortion at nucleosome edges occurs in remodeler–nucleosome or TF–nucleosome complexes (Dodonova et al, 2020; Farnung et al, 2017; Guan et al, 2023; Michael et al, 2020; Willhoft et al, 2018), which can result in increased nuclease sensitivity and partial protection during MNase digestion (Brahma and Henikoff, 2019; Meers et al, 2019). Importantly, these structures retain the histone H3–H4 tetramer at the dyad axis, allowing efficient cleavage by chemical methods (Brahma and Henikoff, 2020), thus offering an explanation for the presence of fragile nucleosomes at NDRs and reconciling discrepancies between MNase and chemical maps. Like histone modifications and

variants, subnucleosomes now emerge as a distinct epigenetic feature of active chromatin (Brahma and Henikoff, 2020), contributing to our understanding of how dynamic nucleosome structures regulate gene expression and chromatin architecture.

Another major advantage of chemical mapping is its unparalleled resolution, enabling human nucleosome maps with single-base-pair precision. This high resolution allows precise measurement of genome-wide nucleosome linker lengths, a critical determinant of higher-order chromatin structure (Correll et al, 2012; Grigoryev, 2018; Wang et al, 2008). Across species, from yeast to mouse to human, chemical nucleosome maps consistently reveal a prevalent $10n + 5$ bp linker pattern, particularly in the short linker range (Brogaard et al, 2012; Moyle-Heyrman et al, 2013; Voong et al, 2016), underscoring a conserved aspect of chromatin organization (Grigoryev, 2018). In this work, we observed distinct linker length distributions between interphase and metaphase chromosomes, indicating global reorganization of nucleosome arrays during the cell cycle. Recent studies suggest that intra-polymer homotypic interactions are crucial for forming chromatin loops, especially those involving promoter–enhancer interactions (Misteli, 2020; Vermunt et al, 2019). Phase-separated condensates have been shown to promote the formation and stabilization of such loops at transcription hubs (Khanna et al, 2019; Misteli, 2020). Intriguingly, recent work indicates that nucleosome-driven phase separation preferentially favors a $10n + 5$ bp spacing pattern and disfavors spacing at exact $10n$ intervals (Gibson et al, 2019). Our chemical mapping data show that interphase linker lengths are significantly shortened near promoters and enhancers, diverging from genome-wide averages and deviating further from their metaphase counterparts, yet still following the $10n + 5$ bp rule. This suggests that modulation of linker length may play a role in the formation of transcriptional condensates during interphase. Supporting this idea, shorter $10n + 5$ bp linker lengths have been shown to enhance the density of phase-separated chromatin condensates (Gibson et al, 2019).

Perhaps the most direct advantage of high-resolution chemical maps is the ability to investigate how DNA sequence features influence dynamic nucleosome positioning. Advances in high-throughput Loop-seq, along with computational tools like DNAcycP and DNAcycP2, have enabled genome-wide computation of intrinsic DNA cyclizability scores (C-scores) (Basu et al, 2021; Kendall et al, 2025; Li et al, 2022). By combining these C-scores with chemically determined dyad positioning data, we examined how DNA mechanics guide nucleosome positioning during mitosis. We find that in mitotic cells, nucleosomes tend to center on the most bendable DNA sequences to minimize energy, consistent with known DNA–histone interaction strengths documented in structural and biophysical studies (Brower-Toland et al, 2002; Luger et al, 1997; Thåström et al, 2004). Strong interactions occur within ±40 bp of the dyad, while contacts further from the center are weaker. Surprisingly, in interphase cells, many nucleosomes preferentially position high-cyclizability sequences at the entry or exit sides of the nucleosome rather than at the dyad. This appears to contradict the expected biophysical behavior of canonical nucleosomes. However, closer analysis shows that these positions are enriched in euchromatic regulatory regions, such as promoters, enhancers, and insulators, where nucleosome arrays are dynamically remodeled during interphase. This dynamic behavior leads to

variations in internucleosome spacing and the formation of subnucleosomal structures. To accommodate transcription and regulatory activity, evolution may have exploited DNA's intrinsic flexibility to enable alternative nucleosome positioning in actively transcribing regions. Importantly, nucleosomes do not act alone. Their positioning is influenced by the coordinated activity of chromatin remodelers and transcription factors (Ahmad et al, 2024; Bulyk et al, 2023). Our findings suggest that mitotic chromosomes reflect a ground state of nucleosome positioning, primarily determined by DNA sequence, whereas interphase chromatin adopts an adaptive, transcriptionally responsive configuration. Together, our study presents a framework for understanding how DNA mechanics and chromatin dynamics jointly govern genome accessibility and compaction through local nucleosome repositioning during the cell cycle.

Finally, we acknowledge certain limitations of the current study. To achieve efficient substitution of endogenous histone H4 with the mutant H4S47C, we used PiggyBac transposons to introduce the H4S47C transgene together with an shRNA targeting wild-type H4. This strategy successfully enabled genome-wide chemical mapping but uncouples H4 expression from its native cell-cycle regulation (Talbert and Henikoff, 2021). Although all H4 isoforms share an identical polypeptide sequence, this decoupling may lead to biased incorporation of H3–H4 dimers that are normally regulated in a cell-cycle–dependent manner (Skene and Henikoff, 2013; Talbert and Henikoff, 2021). Additionally, our interphase maps were generated from asynchronous populations enriched in G1, likely underrepresenting S phase and replicating loci. Future work combining chemical mapping with replication-specific approaches such as MINCE-seq (Ramachandran and Henikoff, 2016) could further elucidate how DNA mechanical properties influence nucleosome dynamics during replication.

# Methods

**Reagents and tools table**

| Reagent/resource | Reference or source | Identifier or catalog number |
|---|---|---|
| **Experimental models** | | |
| Parental HeLa S3 cells | ATCC | CCL-2.2 |
| HeLa S3 Clone 1-2 | This study | N/A |
| HeLa S3 Clone 2 | This study | N/A |
| **Recombinant DNA** | | |
| PB-CAG-H4S47C::PGK-Hygro | Voong | N/A |
| PB-mU6-huH4-sh1::PGK-Puro | This study | N/A |
| **Antibodies** | | |
| Histone H4 antibody | Proteintech | 16047-1-AP |
| Beta-tubulin antibody | DSHB | E7/ AB_2315513 |
| Goat anti-mouse IgG(H/L) HRP | Bio-Rad | 5178-2504 |
| Goat anti-rabbit IgG(H/L) HRP | Bio-Rad | 5196-2504 |
| **Oligonucleotides and other sequence-based reagents** | | |
| 3727.hH4-sh1 sense: tttGGCCTCATCTACGAGGAG AattaaGTtTCCTCGTAGATGAGGCCttttttC | This study | N/A |

| Reagent/resource | Reference or source | Identifier or catalog number |
|---|---|---|
| 3728.hH4-sh1 antisense: tcgaGaaaaaGGCCTCATCTACG AGGAaACttaatTCTCCTCGTAGATGAGGC | This study | N/A |
| **Chemicals, Enzymes and other reagents** | | |
| Fetal bovine serum (FBS) | Gemini Bioproducts | S12450 |
| Trypsin-EDTA | Thermo Fisher | 25200056 |
| DMEM | Thermo Fisher | SH30243FS |
| DPBS | Thermo Fisher | SH30028.LS |
| Penicillin-Streptomycin-Glutamine | Thermo Fisher | 10378016 |
| FuGene HD | Promega | E2311 |
| Thymidine | Thermo Fisher | AAA1149306 |
| Nocodazole | Sigma | SML1665-1ml |
| DNATerminator® End Repair Kit | Lucigen | 40035-2 |
| Klenow DNA Polymerase I | NEB | M0210S |
| Micrococcal nuclease | Sigma | N3755-500UN |
| 3-mercaptopropionic acid | Sigma | M5801-100G |
| Neocuproine | Sigma | N1501-1G |
| Copper(II) chloride dihydrate | Sigma | C3279-100G |
| L-α-Lysophosphatidylcholine from egg yolk | Sigma | L4129-25MG |
| Hydrogen peroxide solution | Sigma | H1009-100ML |
| N-(1,10-phenanthrolin-5-yl) iodoacetamide | Biotum | 92015 |
| Spermine | Thermo Fisher | AAJ6306003 |
| Spermidine | Thermo Fisher | AAA1909603 |
| Proteinase K | Promega | MC5005 |
| Phenol/chloroform/isoamyl alcohol | Thermo Fisher | BP1752I-400 |
| Buffer PB | Qiagen | 19066 |
| AMPure XP Reagent | Beckman Coulter | A63880 |
| NUSIEVE 3:1 agarose | Thermo Fisher | BMA50091 |
| TRIzol Reagent | Thermo Fisher | 15596018 |
| **Software** | | |
| numap R package 0.1.4 | Bioconductor | N/A |
| DNAcycP2 | Bioconductor | N/A |
| STAR/2.7.5 | Dobin et al, 2013 | N/A |
| samtools/1.6 | Danecek et al, 2021 | N/A |
| Adobe illustrator 2025 | adobe.com | N/A |
| Rstudio 2024.12.1 Build 563 | rstudio.com | N/A |
| R 4.5.0 | r-project.org | N/A |

| Reagent/resource | Reference or source | Identifier or catalog number |
|---|---|---|
| **Other** | | |
| Histone Purification Mini Kit | Active Motif | 40026 |
| HiSpeed Plasmid Midi Kit | Qiagen | 12643 |
| Qubit™ dsDNA HS Assay Kit | Thermo Fisher | Q32854 |
| Genejet gel extraction kit | Thermo Fisher | FERK0692 |
| NEBNext® Ultra™ II DNA Library Prep Kit | NEB | E7103S |

## Cell culture and synchronization

HeLa S3 derived cell lines were cultured in high glucose Dulbecco's modified Eagle's (DMEM) medium supplemented with 10% fetal bovine serum (FBS), 1% (v/v) Penicillin/Streptomycin, 1% (v/v) glutamine, and 1 mM sodium pyruvate. For synchronization of HeLa S3 cells to prometaphase, $2 \times 10^6$ cells were seeded per 10-cm dish one day prior to treatment. Cells were first synchronized at early S phase using a double thymidine block (2 mM) over 2 days. After release into fresh medium for 5 h, cells were treated with nocodazole (final concentration: 100 ng/mL) for 8 h to arrest them in mitosis. Mitotic cells were then collected by mechanical shake-off followed by centrifugation.

## Generation of H4S47C-expressing HeLa S3 cell lines

Parental human HeLa S3 cells (ATCC CCL-2.2) were recently authenticated and tested for mycoplasma contamination and was used to generate H4S47C-expressing cell lines for chemical mapping studies. A modified PiggyBac (PB) transgenic strategy, previously described (Voong et al, 2016), was employed, incorporating an shRNA designed against human H4 cDNA isoforms (see Fig. EV1A). Briefly, HeLa S3 cells were co-transfected with the RNAi-resistant H4S47C expression vector PB-CAG-H4S47C::PGK-Hygro, the shRNA construct PB-mU6-huH4-sh1::PGK-Puro, and the transposase expression vector CAG-PBase, using the FuGENE HD transfection reagent (Promega). Stable cell lines were selected sequentially with puromycin and hygromycin. Individual clones were isolated, subcloned, and expanded for subsequent chemical cleavage assays. Of these, the HeLa S3 stable lines "clone 1-2" and "clone 2" were selected for chemical mapping in this study.

## Cell cycle analysis by flow cytometry

Cell cycle profiles of HeLa S3 cell lines used in chemical mapping studies were monitored by flow cytometry (FACScan analysis). Briefly, cells at various cell cycle stages were fixed in 70% ethanol at $-20\,°C$ for at least 2 h, then washed and resuspended in staining buffer containing propidium iodide (PI, 50 μg/mL) and RNase A (100 μg/mL) in PBS. RNase A treatment was included to degrade RNA and ensure that PI staining reflected DNA content exclusively. After incubation at 37 °C for 30 min in the dark, samples were analyzed on a BD LSRFortessa™ Cell Analyzer at the Northwestern University RHLCCC Flow Cytometry Facility to determine DNA content and assess cell cycle distribution.

## Total histone purification and H4 detection

Core histones were extracted from wild-type (WT) and H4S47C-expressing HeLa S3 cells using the Histone Purification Kit (Active Motif), following the manufacturer's instructions. Extracted histones or total cell lysates were separated on a 15% SDS-PAGE gel and visualized either by Coomassie Blue staining or by western blotting using a rabbit anti-histone H4 antibody (Proteintech, 16047-1-AP).

## RNA purification and RNA-seq

Total RNA was extracted from cultured HeLa S3 derivative cells using TRIzol Reagent (Life Technologies) according to the manufacturer's protocol. For RNA-seq analysis, total RNA samples from different cell cycle phases were used to generate RNA-seq libraries with the TruSeq Stranded Total RNA Library Prep Kit (Illumina). Sequencing was subsequently performed on a HiSeq 4000 or NovaSeq 6000 platform at the NUSeq Core Facility, Northwestern University.

## Chemical mapping in interphase and mitotic HeLa S3 cells

To map nucleosome positions in interphase and metaphase cells, we performed chemical cleavage of nucleosomal DNA with modifications based on previously described methods (Voong et al, 2016). Briefly, "clone 1-2" or "clone 2" H4S47C-expressing HeLa S3 cells were harvested either by trypsinization or by shake-off as a cell suspension. Cells were washed with PBS, fixed with 0.5% formaldehyde for 5 min, neutralized with glycine, rinsed with PBS, and incubated in permeabilization buffer (150 mM sucrose, 80 mM KCl, 35 mM HEPES pH 7.4, 5 mM $K_2HPO_4$, and 5 mM $MgCl_2$). Permeabilization was carried out using L-α-lysophosphatidylcholine at a final concentration of 150 μg/mL for 5 min. Cells were then washed with labeling buffer (150 mM sucrose, 10 mM Tris-HCl, pH 7.4, 15 mM NaCl, 60 mM KCl, 5 mM $MgCl_2$, 0.01% NP-40, 0.5 mM spermidine, and 0.15 mM spermine) and incubated for 2 h with 1.4 mM N-(1,10-phenanthro-lin-5-yl)iodoacetamide (Biotium). Following washes with mapping buffer (150 mM sucrose, 50 mM Tris-HCl pH 7.5, 2.5 mM NaCl, 60 mM KCl, 5 mM $MgCl_2$, 0.01% NP-40, 0.5 mM spermidine, and 0.15 mM spermine), cells were treated with 0.15 mM $CuCl_2$ in mapping buffer for 2 min. Unbound $CuCl_2$ was removed by washing, and hydroxyl radical cleavage was initiated by adding 6 mM 3-mercaptopropanoic acid (Sigma) and 6 mM $H_2O_2$ for 2 min. The reaction was quenched with 2.8 mM neocuproine (Sigma). Cleaved genomic DNA was extracted from rinsed cell pellets via proteinase K digestion, followed by phenol/chloroform extraction and ethanol precipitation. Purified DNA was resolved on a 1.8% NuSieve 3:1 (Lonza) agarose gel. DNA fragments (~100–300 bp) were gel-purified and used for paired-end sequencing with the Illumina TruSeq DNA PCR-Free Kit on a HiSeq 4000 or NovaSeq 6000 platform at the NUSeq Core Facility, Northwestern University.

## MNase mapping in interphase and mitotic H4S47C-expressing HeLa S3 cells

MNase digestion of nucleosomal DNA from interphase and metaphase H4S47C-expressing HeLa S3 cells (clone 2 or clone 1-2) was performed as described previously (Voong et al, 2016). Briefly, cells were harvested by trypsinization or mechanical detachment, resuspended in PBS, and fixed with 0.5% formaldehyde for 5 min at room temperature. Following quenching with glycine and subsequent washes, cells were permeabilized in MNase buffer (150 mM sucrose, 50 mM Tris-HCl, pH 7.4, 50 mM NaCl, 0.5 mM $MgCl_2$, 2 mM $CaCl_2$, 0.15 mM spermine, 0.5 mM spermidine) supplemented with 150 μg/mL L-α-lysophosphatidylcholine for 5 min. Permeabilized cells were then incubated with MNase at a final concentration of 100 U/mL at 37 °C for 10 min. Reactions were terminated by the addition of Stop Buffer (10 mM Tris-HCl, pH 7.4, 100 mM NaCl, 10 mM EDTA, and 1% SDS), and digested genomic DNA was purified following overnight proteinase K treatment. Mononucleosomal DNA fragments were resolved on a 1.5% agarose gel, gel-purified, and prepared for sequencing using the TruSeq DNA PCR-Free Kit on an Illumina sequencing platform.

## Sequencing and read alignment

All DNA sequencing libraries were prepared for paired-end sequencing using standard Illumina protocols. For chemical mapping, two biological replicates were sequenced for each condition—interphase and metaphase. Sequencing reads were aligned to the Ensembl human genome assembly GRCh38.p13 (GCA_000001405.28, Release 108) using the STAR aligner, and only uniquely mapped reads were retained for calculating chemical cleavage frequencies. Similarly, for MNase mapping, only uniquely mapped reads were used. In total, approximately 3.17 and 2.2 billion uniquely mapped reads were obtained for the interphase and metaphase chemical maps in clone 2, respectively, and 1.7 and 1.1 billion reads for clone 1-2. For the MNase maps in clone 2, 262 million and 222 million reads were mapped for the interphase and metaphase samples, respectively.

## RNA-seq data analysis

RNA-seq data from various interphase HeLa S3 cell lines were aligned to the human genome assembly GRCh38.p13 using the STAR aligner. Only uniquely aligned reads were used to quantify gene-level expression, measured as fragments per kilobase of transcript per million mapped reads (FPKM). Of the 19,166 protein-coding genes annotated, 15,887 showed non-zero read counts.

## NCP score calculation and unique/redundant nucleosome definition in chemical maps

We followed the approach described by Brogaard et al (Brogaard et al, 2012) to identify cleavage patterns around nucleosome dyads and used the R package "numap" (https://github.com/jipingw/numap), implementing a deconvolution algorithm, to compute nucleosome center positioning (NCP) scores at every genomic location (Xi et al, 2014).

Unique nucleosome positions were defined using a greedy algorithm based on the magnitude of NCP scores, with a minimum spacing of ≥120 bp between adjacent dyads (Brogaard et al, 2012). Briefly, the genomic location with the highest NCP score was assigned as the first nucleosome center. The next highest-scoring position located at least ±120 bp away from previously assigned centers was selected as the next nucleosome center, and the process

continued iteratively. From the resulting set of nucleosome positions, the top 90% (based on NCP score) were retained to generate the unique nucleosome maps, representing the most prominent dyad positions. For both interphase and metaphase, ~14.9 million unique nucleosomes were identified.

To define redundant nucleosome maps, we included all genomic locations with NCP scores greater than or equal to the minimum NCP score found in the unique map. This resulted in 126 million and 125 million redundant nucleosome positions for interphase and metaphase, respectively.

## Nucleosome occupancy calculation

Throughout this study, nucleosome occupancy is defined as the center-weighted nucleosome occupancy, calculated from the redundant nucleosome map as previously described (Voong et al, 2016). For the chemical map, at a given genomic location, the NCP score was kept unchanged if a redundant nucleosome was defined; otherwise, the score was set to 0. The center-weighted nucleosome occupancy score is defined as the weighted average of NCP scores from redundant nucleosomes, with weights proportional to a Gaussian density centered at 0 and a standard deviation 20 bp, decaying symmetrically from the center.

For MNase maps, the same calculation is applied, with NCP scores replaced by the number of MNase-seq reads centered at the genomic position.

## Local correlation and enrichment analysis

We calculated the Pearson correlation of nucleosome occupancy between interphase and metaphase across the genome using a sliding window of 501 bp with a 1 bp step size. These local correlation values were used in downstream analyses to assess the consistency of nucleosome occupancy between interphase and metaphase around TSSs, CTCF-binding sites, and active enhancers.

For enrichment analysis, we divided the genome into 3,029,672 consecutive, non-overlapping 1000 bp bins. Pearson correlation coefficients were calculated for each bin to evaluate nucleosome occupancy similarity between interphase and metaphase. A total of 2,751,637 bins had valid correlation values after excluding unmappable regions or bins lacking nucleosome signal in either condition. Each bin was then classified as either low or high correlation based on a threshold of 0.15, resulting in 127,136 low-correlation bins.

We defined Regions of Interest (ROI) as bins that either (i) belong to specific chromatin states of interest or (ii) overlap with target genomic features such as TSSs, CTCF-binding sites, or active enhancers. Bins that did not meet these criteria were designated as Background (BG). For chromatin state-based classification, a bin was considered an ROI if the majority of its sequence (≥500 bp) was annotated with a specific chromatin state according to ChromHMM predictions; otherwise, it was assigned to BG.

To quantify enrichment, we constructed $2 \times 2$ contingency tables for each ROI–BG comparison based on the number of low- and high-correlation bins (see Fig. 2C). Enrichment was measured using the log odds ratio, and statistical significance was assessed by a Chi-square test (df = 1). Corresponding $p$ values are reported in Appendix Table S1.

## Test for shorter linker length enrichment in chromatin states

To assess whether short linker lengths (<30 bp) are more enriched in interphase than in metaphase within specific chromatin states or genomic features, we performed a Wald test on differences in log odds ratios. For each chromatin state group, we constructed a $2 \times 2$ contingency table with rows representing the two cell cycle conditions (interphase and metaphase) and columns corresponding to linker length categories (<30 bp vs. ≥30 bp). Odds ratios for short linker enrichment in interphase relative to metaphase were calculated for each group, and the Wald test was used to determine whether the log odds ratio in one group was significantly greater than in another. For the chromatin states shown in Fig. 2E, we computed odds ratios for the following groups: E1–E4, E5/E6, E9–E11, E13, and E18, and tested whether the enrichment observed in E1–E4 or E9–E11 was significantly greater than in the others. For TSSs, enhancers, and CTCF-binding sites, we calculated odds ratios for interphase versus metaphase within the regions of interest and compared them to genome-wide background regions, excluding those elements. Results are provided in Appendix Tables S2 and S3.

## CTCF data analysis

CTCF ChIP-seq data were obtained from the ENCODE Project (Ram et al, 2011; Data ref: Gene Expression Omnibus GSM733785, 2011), with the corresponding control dataset (Data ref: Gene Expression Omnibus GSM733659, 2011). Peak calling was performed using MACS3 (Zhang et al, 2008), resulting in the identification of 74,550 CTCF peaks with fold enrichment values ranging from 1.422 to 85.5. Due to the inherent resolution limitations of ChIP-seq, peak positions were further refined using a position weight matrix (PWM) for the CTCF motif from JASPAR (MA0139.1, https://jaspar.genereg.net/matrix/MA0139.1/). This 19-nucleotide motif model was used to scan both strands within each MACS3-called peak window. The position with the highest motif match score within each window was defined as the refined CTCF center.

## Transcription start site (TSS) analysis

We compiled a set of 19,166 transcription start sites (TSSs) and transcription termination sites (TTSs) for protein-coding genes based on the annotation of the human genome (GRCh38.p13). For each protein-coding gene, the transcript annotated as ensembl_havana in the transcript_source field of the GTF file was selected as the representative isoform. If no ensembl_havana annotation was available, the transcript with the longest coding region was chosen instead.

## Intron–exon and exon–intron boundaries analysis

We compiled a total of 164,916 non-terminal exons from the 19,166 protein-coding genes highlighted in Fig. 3. Exon center positions were defined as the midpoint of each exon; for exons with an even number of nucleotides, the leftmost of the two central bases was selected. Intron–exon boundaries were defined as the 5′ start positions of exons, while exon–intron boundaries were

defined as the 3′ end positions of exons. NET-seq data used in this analysis were obtained from a previously published dataset (GSE61332) (Mayer et al, 2015; Data ref: Gene Expression Omnibus GSE61332, 2015).

## Blinding statement

No applicable blinding was done in this study.

## Data availability

The datasets and computer code produced in this study are available in the following databases: - Chemical mapping for clone 2: gene expression omnibus accession GSE264401 (https://www.ncbi.nlm.nih.gov/geo/query/acc.cgi?acc=GSE264401). - MNase-seq for clone 2: gene expression omnibus accession GSE264399 (https://www.ncbi.nlm.nih.gov/geo/query/acc.cgi?acc=GSE264399). RNA-seq for clone 2: gene expression omnibus accession GSE264400 (https://www.ncbi.nlm.nih.gov/geo/query/acc.cgi?acc=GSE264400). Chemical mapping for clone 1-2: gene expression omnibus accession GSE296340 (https://www.ncbi.nlm.nih.gov/geo/query/acc.cgi?acc=GSE296340). MNase-seq for clone 1-2: gene expression omnibus accession GSE296337 (https://www.ncbi.nlm.nih.gov/geo/query/acc.cgi?acc=GSE296337). RNA-seq for clone 1-2: gene expression omnibus accession GSE296336 (https://www.ncbi.nlm.nih.gov/geo/query/acc.cgi?acc=GSE296336). DNAcyP2-predicted DNA cyclizability scores: Zenodo repository (https://zenodo.org/records/17393602). MNase and chemical mapping processing scripts: Numap R package on GitHub (https://github.com/jipingw/numap).

The source data of this paper are collected in the following database record: biostudies:S-SCDT-10_1038-S44320-026-00192-y.

## Peer review information

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

## Acknowledgements

We thank the Northwestern NUSeq Core Facility for advice and Illumina sequencing services. YT was supported in part by the CMBD training grant NIH T32 GM08061. J-PW and XW were recipients of the NuSeq Illumina Pilot Program. The work was partially supported by NSF-Simons Center for Quantitative Biology [SFARI 597491-RWC; DMS-1764421] to J-PW and XW, and by NIH grants R01GM120307 and R01GM149076 to XW.

## Author contributions

**Keren Li**: Resources; Data curation; Software; Formal analysis; Investigation; Visualization; Methodology; Writing—original draft; Writing—review and editing. **Irem Unlu**: Conceptualization; Data curation; Formal analysis; Investigation; Visualization; Methodology. **Yiren Tu**: Resources; Formal analysis; Validation; Investigation; Visualization; Writing—review and editing. **Lilien N Voong**: Methodology. **Yanyan Lu**: Data curation; Investigation. **Brody Kendall**: Resources; Software. **Xiaotian Ma**: Software; Investigation. **Sin Lei Pui**: Data curation; Methodology. **Meng Tao**: Data curation; Visualization. **Ji-Ping Wang**: Conceptualization; Resources; Data curation; Software; Formal analysis; Supervision; Funding acquisition; Investigation; Visualization; Writing—original draft; Project administration; Writing—review and editing. **Xiaozhong Wang**: Conceptualization; Data curation; Formal analysis; Supervision; Funding acquisition; Validation; Investigation; Visualization; Methodology; Writing—original draft; Project administration; Writing—review and editing.

Source data underlying figure panels in this paper may have individual authorship assigned. Where available, figure panel/source data authorship is listed in the following database record: biostudies:S-SCDT-10_1038-S44320-026-00192-y.

## Disclosure and competing interests statement

The authors declare no competing interests.

# Expanded View Figures

**Figure EV1.   Development of chemical nucleosome mapping in HeLa S3 cells.**

(A) Design of a common RNAi target sequence shared by all human histone H4 isoforms. The siRNA antisense strand is designed to bind H4 mRNA via Watson–Crick or wobble base pairing. (B) Comparison of histone levels in parental and H4S47C-expressing HeLa S3 cell lines. Top: SDS-PAGE analysis of total histones purified from parental, clone 1-2, and clone 2 cells. Bottom: Western blot analysis of H4 levels in the same histone samples. (C) Growth curve showing that H4S47C-expressing HeLa S3 cells proliferate at the same rate as wild-type (WT) cells. Data represent the mean of biological triplicates; error bars indicate SEM. (D) Scatter plot of $\log_2$(FPKM) values comparing transcriptomes of clone 1-2 and clone 2. (E) FACScan analysis of cell cycle distribution in asynchronous (interphase) and synchronized prometaphase HeLa S3 cells. (F) Western blot analysis of H4 levels in total cell lysates from interphase and metaphase HeLa S3 cell lines. (G) DNA laddering pattern from chemically cleaved nucleosomes in interphase and metaphase clone 2 cells.

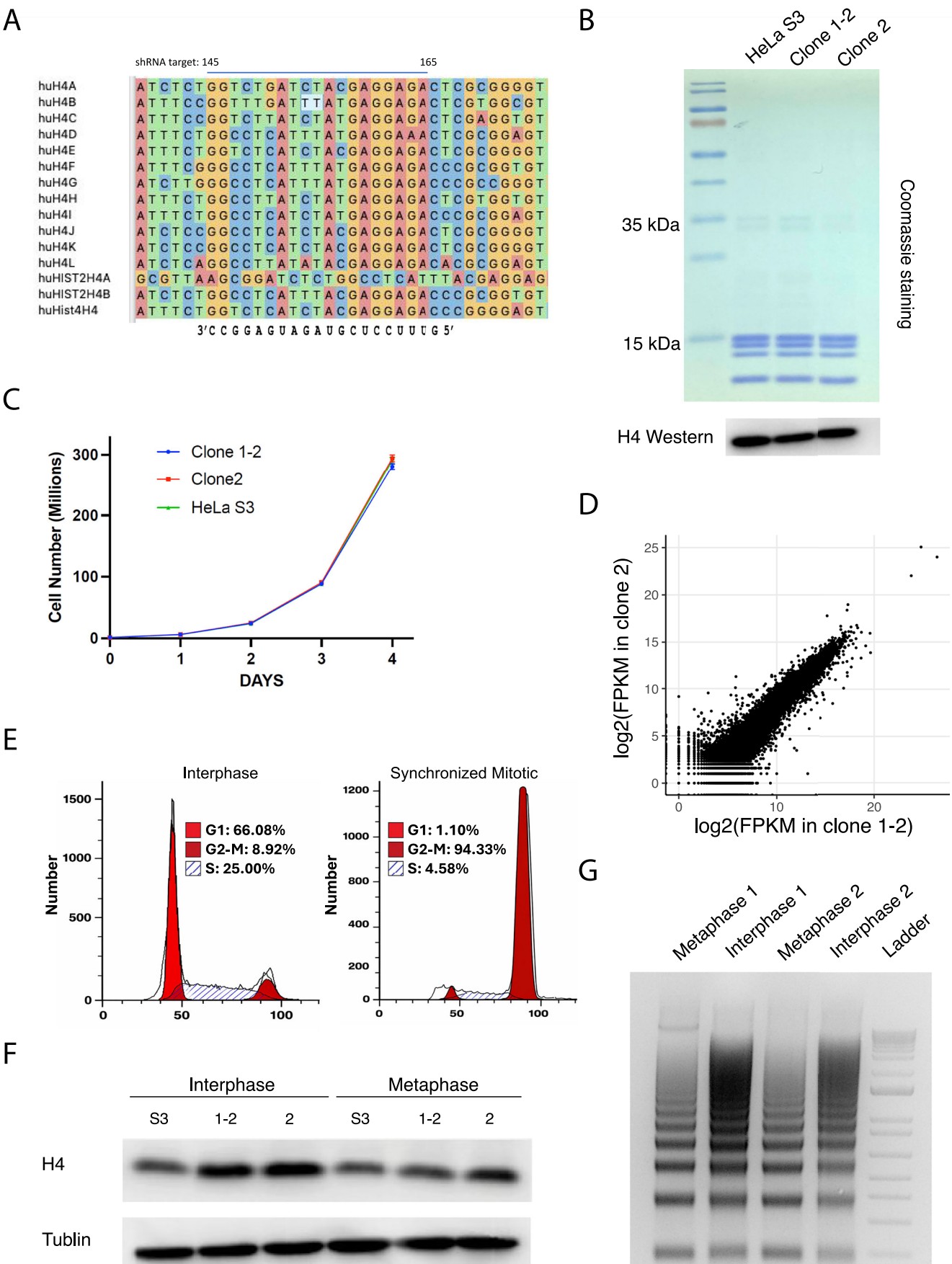

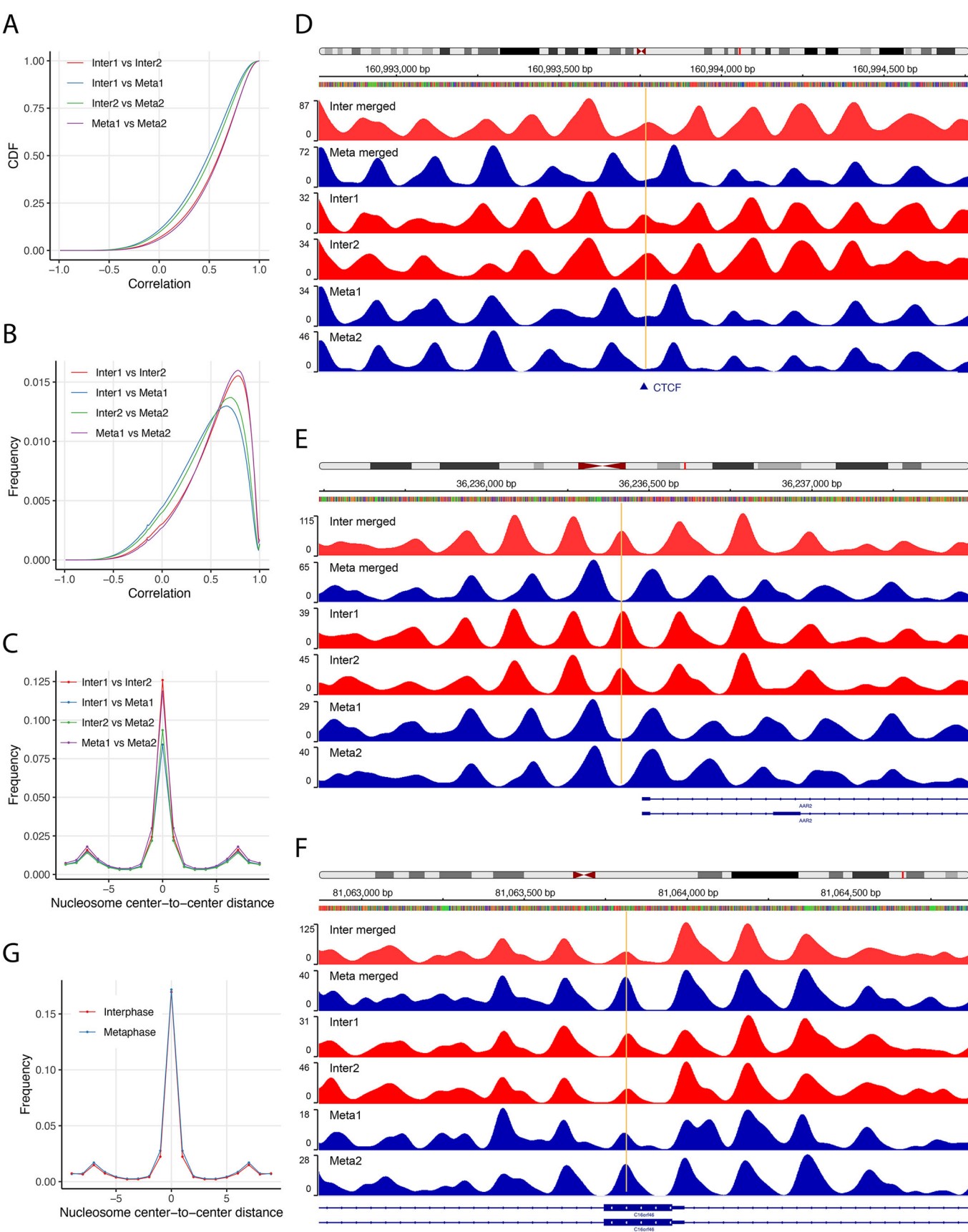

◀ **Figure EV2. Reproducibility of chemical nucleosome maps in H4S47C-expressing HeLa S3 cells.**

(A) Empirical cumulative distribution function (ECDF) plots showing genome-wide local Pearson correlation of nucleosome occupancy scores between biological replicates of clone-2 H4S47C-expressing cells. Comparisons include: Interphase 1 vs. Interphase 2, Metaphase 1 vs. Metaphase 2, Interphase 1 vs. Metaphase 1, and Interphase 2 vs. Metaphase 2. Correlations were computed using a 501-bp sliding window with 1-bp step size. (B) Frequency distribution of local correlation values for the same replicate comparisons shown in (A). (C) Distribution of center-to-center distances between unique nucleosomes identified in each pairwise comparison from (A). (D–F) Representative genomic loci illustrating nucleosome occupancy scores before and after merging clone-2 biological replicates. (G) Frequency plot of center-to-center distances between unique nucleosomes in independent clones. Shown are comparisons between clone 1-2 and clone 2 for both interphase and metaphase chemical nucleosome maps.

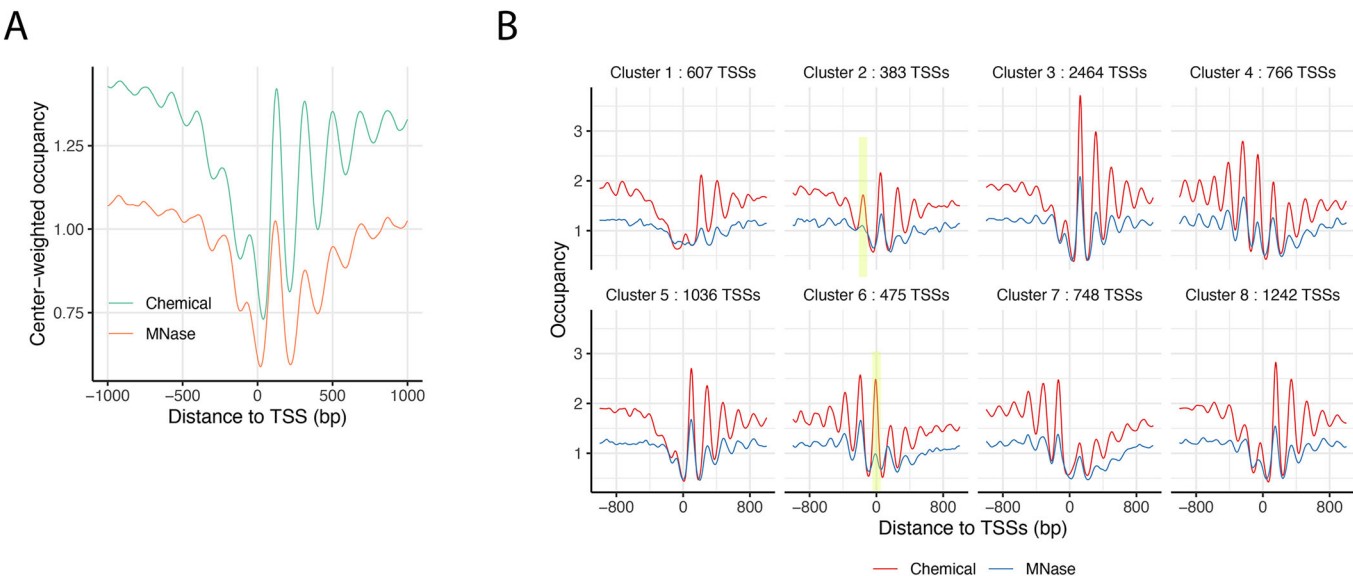

**Figure EV3.  Nucleosome occupancy over TSSs during interphase.**

(A) Average interphase nucleosome occupancy profiles from chemical (green) and MNase (red) maps across the TSSs of 19,166 protein-coding genes. (B) A subset of 7,944 TSSs, representing the top 50% of expressed genes, was grouped into eight clusters using K-means clustering based on interphase chemical occupancy scores in the region spanning –150 to +250 bp relative to the TSS (same data as Fig. 3D). Shown are the interphase occupancy profiles from the chemical and MNase maps for each cluster. Notably, clusters 2 and 6 exhibit a well-positioned –1 nucleosome in the chemical map that is not detected in the MNase map.

A

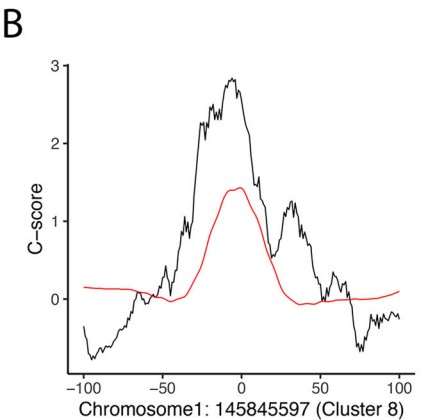

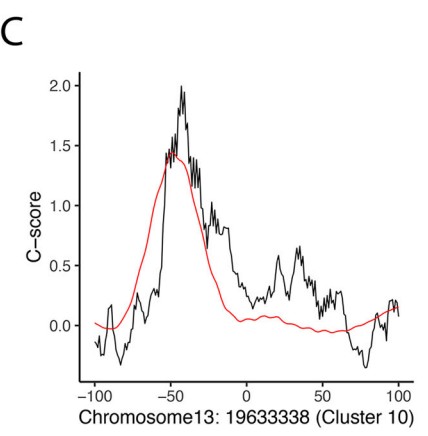

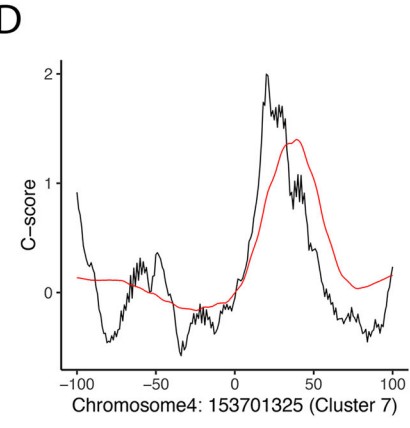

**Figure EV4. Distribution of DNA C-scores within nucleosome regions.**

(**A**) Unique nucleosomes near transcription start sites (TSSs) in interphase were clustered into 14 groups using a K-means algorithm, revealing distinct patterns of intrinsic DNA cyclizability (C-scores) within the nucleosome region. (**B–D**) C-score distributions for three representative nucleosomes (black). Red curves indicate the average profile for the corresponding cluster.

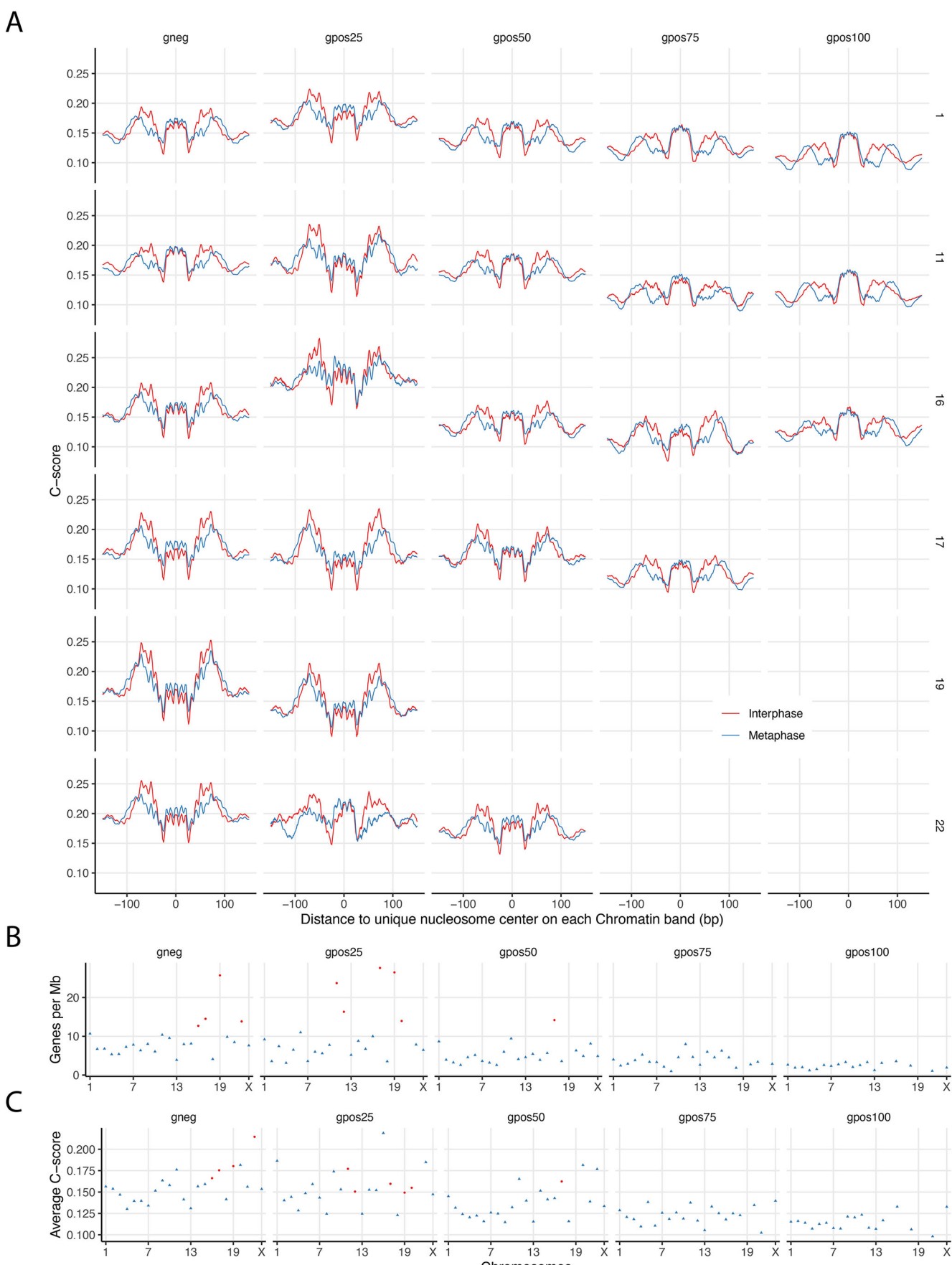

◀ **Figure EV5. Relationship between gene density and nucleosomal C-score patterns.**

(A) Representative examples of nucleosomal C-score distributions across various G-band regions on different chromosomes. (B) Gene density (genes per Mb) across G-band regions on individual chromosomes. Regions with gene density >11 genes/Mb are marked as red dots; all others are shown as blue triangles. These regions correspond to those in (A) and exhibit significantly higher C-scores in the nucleosome shoulder compared to the dyad. (C) Distribution plot showing average C-score values within different G-band regions. Regions are colored according to gene density, as in (B).

