## [Peer Review File · Molecular Systems Biology]

Differential nucleosome organization in human interphase and metaphase chromosomes

Keren Li, Irem Unlu, Yiren Tu, Lilien Voong, Yanyan Lu, Brody Kendall, Xiaotian Ma, Sin Lei Pui, Meng Tao, Ji-Ping Wang, and Xiaozhong Wang

Corresponding author(s): Xiaozhong Wang (awang@northwestern.edu) , Xiaozhong Wang (awang@northwestern.edu), Ji-Ping Wang (jzwang@northwestern.edu)

Review Timeline:

Transfer Date:	21st Jul 25
Editorial Decision:	11th Sep 25
Revision Received:	21st Oct 25
Editorial Decision:	9th Dec 25
Revision Received:	2nd Jan 26
Accepted:	13th Jan 26

Editor: Poonam Bheda

Transaction Report: The first round of review of this manuscript was performed at another journal.

Response to review

“While we cannot continue with this manuscript at our journal, I have taken time to discuss the study with our colleagues at the Molecular Systems Biology. As you may know, Molecular Systems Biology is a non-profit, fully open-access journal published by EMBO Press and offers rapid and transparent peer review with a single major revision round policy. I am glad to say that the handling editor Poonam Bheda would be happy to send your manuscript back for re-review at Molecular Systems Biology. However, please note that in line with comments from Reviewer 2, the editors at Molecular Systems Biology would require that an additional PiggyBac clone or cell line be analyzed to ensure the robustness of the data as a resource. If you are interested in this option, please submit the manuscript to Molecular Systems Biology via the transfer link below. Please feel free to contact Poonam at p.bheda@molsystbiol.org if you have any questions about the transfer, Molecular Systems Biology, or its policies. Please note that no reformatting of the manuscript is necessary for the transfer”.

I. Summary of major changes in revision in response to review

Following initial review at *[journal name redacted]* our manuscript was transferred to *Molecular Systems Biology*. We thank the reviewers and editors for their constructive feedback. Over the past year, we have conducted substantial new experiments to address the reviewers' concerns. The revised manuscript includes three major improvements:

1. New PiggyBac clone data.
We generated and analyzed chemical mapping data from an additional PiggyBac clone. Results from this new clone are presented in the main text, while data from the original clone are included in the Appendix for validation. The two datasets are consistent with each other.
2. Improved DNA cyclizability analysis with DNAcycP2
We developed an enhanced computational tool, DNAcycP2 (PMC11897897), which uses a novel data augmentation approach to improve estimates of intrinsic DNA cyclizability. Since previous submission, all relevant analyses have been reprocessed using DNAcycP2.
3. Biological and technical replicates
To assess reproducibility, we generated and analyzed chemical maps from biological replicates and independent clones. These confirmed the overall consistency of nucleosome positions determined by chemical mapping.

Despite incorporating entirely new chemical map data, we reached the same key conclusions as in the original submission.

II. Point-by-point response

Referee #1

(Remarks to the Author)

Review of the manuscript by Li and colleagues

Li and colleagues initiated a highly interesting study to compare the chromatin architecture of interphase and metaphase chromosomes. Furthermore, they aim to compare the chemical mapping technology with MNase-based chromatin structure analysis.

When I started reviewing the highly interesting topic, I realized that the manuscript did not sufficiently

address or show fundamental questions regarding the quality of the obtained data. Therefore, I would like to ask the authors to clarify the points mentioned before reviewing the novel findings.

The authors described the chemical mapping of chromatin in HeLa S3 cells stably knocked down for H4 and overexpressing the RNAi-resistant H4S47C variant. No data is provided quantifying the remaining endogenous histone H4 levels and the levels of H4S47C.

- Is the total histone level per cell the same for the selected "1-2" clone compared to non-modified HeLa S3 cells? It can be assumed that differences in histone protein concentrations will dramatically affect nucleosome positioning and spacing.

Response: We thank the reviewer for this important critique. In the revised manuscript, we now include protein analyses of H4 and other core histones in H4S47C-expressing clones ("1-2" and "2") as well as in parental HeLa S3 cells. These data demonstrate that total histone levels, including H4, are comparable between the H4S47C clones and the parental HeLa S3 cells (Fig. EV1B).

- Furthermore, histone gene expression is tightly regulated throughout the cell cycle. Does the PB-CAG-H4S47C construct mimic this effect? Do the cells propagate with a "normal" kinetics through the cell cycle? How do growth conditions and cell cycle progression of the "wild-type" HeLa cell and the "1-2" clone compare?

Response: We have now included a comparison of growth curves for two H4S47C-expressing clones and parental HeLa S3 cells. These data show that the H4S47C-expressing clones exhibit normal proliferation rates under standard growth conditions (Fig. EV1C).

- Quantification should be performed on protein level and RNA-seq level, to distinguish and identify the RNAi resistant H4S47C transcript.

Response: We analyzed RNA-seq data from these clones and found that gene expression profiles were similar between them (Fig. EV1D). In clones 1-2 and 2, 76.7 percent and 77.8 percent of total H4 mRNAs, respectively, mapped to the RNAi-resistant H4S47C transgene, confirming its predominant expression.

Why did the authors not exchange some endogenous H4 copies using CRISPR techniques to avoid dose effects? This would have maintained the expression dynamics of the endogenous locus.

Response: The human genome contains 15 distinct histone H4 genes, each present in two copies. These genes are dispersed across the genome, with some organized in clusters interspersed with other genes. Due to this complex genomic arrangement, it is extremely challenging to replace a substantial fraction of H4 genes using CRISPR-based approaches.

Regarding the concerns about histone load in the cell, it is essential to perform the MNase-seq experiments as well in the "1-2" stable clones and not in regular HeLa S3 cells, as described in Material and Methods.

Response: We apologize for the oversight. The Methods section did not clearly state that the HeLa S3 cells used for mapping were from the "1-2" cell line. In the revised manuscript, we have clarified that all MNase-seq data presented in this study were generated using the H4S47C-expressing "1-2" or "2" HeLa S3 cells.

Furthermore, the sequencing depth of the MNase-seq data does not compare to the chemical mapping data. For proper MNase-seq data and nucleosome annotation, the sequencing depth is too low.

Response: We thank the reviewer for this point. Unlike chemical mapping, which requires high cleavage depth at base-pair resolution to accurately resolve dyad positions using deconvolution algorithms, MNase-seq relies on read coverage to estimate nucleosome occupancy. Therefore, MNase-seq does not require comparable read counts to chemical mapping for reliable occupancy patterns. For example, 10 reads at a nucleosome site in MNase-seq would yield a stable coverage signal across the 147 bp region, whereas the same number of chemical mapping reads would be distributed across primary, secondary, and background cleavage sites, making dyad calling less robust at low depth. In the revised manuscript, for the MNase maps in clone 2, 262 million and 222 million reads were mapped for the interphase and metaphase samples, respectively.

The authors performed two replicates of the chemical mapping and the MNase-seq. It is required to compare and quantify the variation between replicates, to allow the detection of significant differences. To provide a better impression of the variation and significance of the experiments, the authors should change the presentation of the genome browser screenshots (Figure 1d-f; S1g-i) not only showing the pooled mean of the replicates. A presentation showing the minimum/maximum intervals of all replicates and cutting sites would be appropriate. Are the differences in the map significant?

Response: We thank the reviewer for this helpful suggestion. We initially pooled the replicates because cleavage counts strongly influence the accuracy of nucleosome mapping. However, we agree that presenting individual replicates provides a clearer view of reproducibility. In the revised manuscript, we included a comparison of individual replicate data and data from independent clones (Fig. EV2, Appendix Supplementary Figures). Specifically addressing the reviewer's concern, we now show genome browser screenshots for both replicate and merged datasets from clone 2 (Fig. EV2D-F). In addition, we compare representative tracks from two independent clones (1-2 and 2) in Fig. 1E-G. Additional statistical analyses quantifying variation between replicates are presented in Fig. EV2A-C.

The authors created two nucleosome maps out of their chemical maps (line 78+). How do the 15 million map and the 125 million nucleosome maps overlap? The human genome has space for only 30 million nucleosomes?! Are the 15 million high-quality nucleosomes in different regions than the supposedly overlapping 125 million positions? In my opinion, a clearer evaluation and description of the basic data is required, before discussing the potential results.

Response: We appreciate the reviewer's comment and recognize that the distinction between the two maps may not have been clearly explained. In the revised manuscript, we clarify this point. Briefly, the 15 million nucleosome map refers to the "unique" set, in which nucleosomes are required to be spaced at least 120 bp apart. This set highlights the most prominent and confidently positioned nucleosomes. The 125 million map is the "redundant" set, which includes all positions with a nucleosome center (NCP) score above a defined threshold (typically, the minimum NCP score found in the unique map), without a spacing constraint. This map captures alternative dyad positions that arise due to nucleosome rotational positioning or heterogeneity across the cell population. These alternative positions often occur at ~10 bp intervals from the primary dyad. Chemical mapping enables us to resolve this positional diversity, which traditional MNase mapping cannot. The inclusion of both maps allows us to analyze nucleosome occupancy and positioning with greater resolution and biological relevance.

Downloading and superficially looking at it, raised many questions (only few questions marked in the

picture)?

XX figure and document is uploaded XXX

Response: We don't understand this comment.

The lines shown are the high confidence position – I suppose the 15 million nucleosome map. But the lines show already a smoothing and statistical analysis of the data. How does the raw data of the individual replicates look? And if the positioning mapping is that unsharp with the high quality map. How bad is it with the 125million map. And how could a significant overlap analysis be performed with the nucleosomal map.

Cleavage sites have to be taken individually and being analyzed. The cleavage sites represent absolute positions of within a cell. In my opinion the authors have to lay the statistical ground work and show the robustness of their analysis.

Response: We believe this comment relates to the earlier point regarding the differences between the 15 million and 125 million nucleosome maps. In the revised manuscript, we included a comparison of individual replicate data and data from independent clones (Fig. EV2, Appendix Fig. S3-S7). We also provided additional analyses demonstrating the statistical significance of differential nucleosome positions and linker length distributions (Appendix Table S1, Table S2, Table S3). Detailed statistical methods are described in the revised Methods section.

I suggest to address the mentioned points prior and then re-submitting the manuscript for evaluation.

Response: We have revised the manuscript to address all points raised by the reviewer and are now resubmitting it for evaluation.

Referee #3

(Remarks to the Author)

The study by Li et al reports chemical mapping of nucleosomes in human (HeLa S3) cells by an analogous approach as the chemical mapping in mESCs reported in ref. 10 by the same groups. In this regard, this new study seems like “more of the same”, just in human rather than mouse cells, and with a bit of a different biological question comparing interphase and metaphase cells.

In both the 2016 and now the 2024 study, the approach relies on chemical cleavage of the DNA backbone by hydroxyl radicals locally generated after incubation with phenanthroline and copper ions at an engineered cysteine residue in histone H4 (H4S47C). This method was developed in the Richmond group in the 90ies (ref. 11) and taken to the genome-wide level by Jon Widom in collaboration with Ji-Ping Wang for both *S. cerevisiae* and *S. pombe* (refs. 9 and 12). Ji-Ping Wang's and Xiaozhong Wang's groups took this approach even further to genome-wide chemical mapping in mouse (ref. 10) and human cells (the here reviewed study). All these chemical mapping studies are great to have and I appreciate all these efforts by these groups. The chemical mapping approach is for sure superior in accuracy (low false positive rate) and resolution (one bp) to the classical MNase-seq approach and allows, for example, the kind of DNA sequence analyses as also reported here. For example, the dinucleotide periodicity patterns (Figs. 1B, S1D,E,F), the linker length histograms (Figs. 2D,E, 3E, 4G, 5B, S3) or the here newly adopted DNA cyclizability profiles (Figs. 6, S6-8) are most valid if derived from chemical mapping data. In this regard, I really appreciate, as repeatedly stated by the authors, that this new 2024 study reports for the first time a genome-wide chemical nucleosome map for human cells. This is a great achievement and resource and offers new biological insights.

Response: We thank the reviewer for their thoughtful and encouraging comments. We greatly

appreciate the recognition of the technical and biological value of chemical nucleosome mapping, as well as the historical context and continuity of this methodological development. As noted, our study builds upon prior work by extending chemical mapping to a genome-wide scale in human cells, enabling high-resolution analysis of nucleosome organization across interphase and metaphase. We are pleased that the reviewer finds the newly introduced analyses, such as linker length distributions and DNA cyclizability profiles, to be robust and insightful. We hope this dataset will serve as a valuable resource for the broader chromatin biology community.

(At the side, the only other group, to my knowledge, that did genome-wide nucleosome mapping so far is the Henikoff group (PMID 29426353). There, they used the H3Q85C mutant, which seems superior to the H4S47C mutant for the reasons discussed in their paper, i.e., clearly defined identification of nucleosomes via the 51 bp fragments (even lower false positive rates) and no problem with mapping nucleosomes next to NDRs, like +1 nucleosomes, where the neighboring nucleosome is missing and the dyad-to-dyad DNA fragments generated with the H4S47C mutant become difficult to detect. I wonder why the authors chose to stick with H4S47C.)

Response: We agree with the reviewer that H3Q85C may offer technical advantages for mapping dyad positions, especially in nucleosome-depleted regions (NDRs). However, we chose to use H4S47C for mammalian cell applications because the H3 family in mammals consists of multiple isoforms (e.g., H3.1, H3.2, H3.3) with distinct amino acid sequences and functional roles. This complexity makes gene replacement approaches challenging, as it would be difficult to maintain the endogenous expression balance among H3 isoforms. In contrast, although there are multiple H4 and H2B genes in the genome, they encode identical proteins, enabling the use of a single synthetic transgene to achieve effective replacement. This rationale underlies our continued use of H4S47C in human cells.

Nonetheless, I have several concerns.

1) Nowadays, it is unacceptable to base a whole study using random PiggyBac integration on one single clone. This was already the case in the 2016 study and now again, even though the authors had no problem generating several clonal cell lines: "Individual clones were picked, subcloned, and expanded for the chemical cleavage reaction. Among these, the HeLa S3 stable cell line "1-2" was selected for chemical mapping in this study." (lines 539-541) Given the stochasticity of the PiggyBac integration and possible side effects, at least two, better even three or more independent clones have to be analyzed.

Response: We thank the reviewer for this important comment. We fully recognize the concern regarding reliance on a single PiggyBac clone. In the revised manuscript, we have included data from an independently generated clone ("clone 2"), which was selected for the main analyses due to its higher sequencing depth. Data from the original "clone 1-2" are presented in the Appendix as supplementary validation. Importantly, results from both clones are highly consistent and support the same conclusions, strengthening the robustness of our findings.

2) The authors merge the sequencing data from both their technical replicates (line 592). In addition to using true biological replicates (see point 1), it would be more informative to analyze the technical replicates individually so that the technical variation becomes apparent, and then average over the technical replicates.

Response: Please also see our response to Reviewer 1's related comment. We agree that analyzing technical replicates individually provides important insights into variation and reproducibility. In the revised manuscript, we generated individual chemical maps for each replicate and quantified the variation between them (Fig. EV2A–C). We also included genome browser screenshots comparing both individual and merged replicate datasets from clone 2 (Fig. EV2D–F). Additionally, representative tracks from two independent clones (1-2 and 2) are shown in Fig. 1E–G. These analyses demonstrate the consistency across replicates and clones.

3) The authors have to use the same cell lines for their MNase-seq mapping as for their chemical mapping. Currently the authors use (line 576) plain HeLa S3 cells without the PiggyBac treatment. This makes their comparisons between the two different nucleosome mapping approaches like comparing apples with oranges.

Response: We apologize for the oversight. The Methods section did not clearly state that the HeLa S3 cells used for mapping were from the "1-2" cell line. In the revised manuscript, we have clarified that all MNase-seq data presented in this study were generated using the H4S47C-expressing "1-2" or "2" HeLa S3 cells.

4) As already the case for the 2016 study (ref. 10), there is a principle problem for chemical mapping in metazoan cells. In contrast to yeasts (refs. 9 and 12), where it is possible to construct strains with the H4S47C mutant allele as the sole histone gene allele in the cell, mouse and human cells contain so many histone H4 gene copies (15 copies in human cells according to Fig. S1A) that these wild-type genes cannot all be removed prior to or concomitant with introduction of the mutant allele. Instead, the authors target the wild-type genes by siRNA and overexpress the mutant allele from the strong CAG promoter and from an unknown number of copies after stochastic (see point 1 above) PiggyBac integration. In contrast to the 2016 study (Fig. S1D, E, F there), Li et al. do not show or write about any controls regarding the expression levels (RNA, protein) of the mutant H4 relative to wild-type H4 in their human cells, nor do we know anything about a change in growth rate.

Response: Please also see our response to Reviewer 1's related comments.

In the revised manuscript, we now include protein analyses of H4 and other core histones in H4S47C-expressing clones ("1-2" and "2") as well as in parental HeLa S3 cells. These data demonstrate that total histone levels, including H4, are comparable between the H4S47C clones and the parental HeLa S3 cells (Fig. EV1B).

We have now included a comparison of growth curves for two H4S47C-expressing clones and parental HeLa S3 cells. These data show that the H4S47C-expressing clones exhibit normal proliferation rates under standard growth conditions (Fig. EV1C).

We analyzed RNA-seq data from these clones and found that gene expression profiles were similar between them (Fig. EV1D). In clones 1-2 and 2, 76.7 percent and 77.8 percent of total H4 mRNAs, respectively, mapped to the RNAi-resistant H4S47C transgene, confirming its predominant expression.

Indeed, it was hard to believe in the 2016 study that knocking down all 13 wild-type histone H4 gene copies and replacing them by one or few copies of the H4S37C gene did not lead to slower growth as S phase should be slowed down if histone synthesis rate is reduced due to lower copy number. As the growth rate did not decrease (Fig. S1F in Voong et al., 2016, Cell) it seems very likely that there was still a rather high expression level of wild-type histone H4 in these cells. This means that the nucleosome

positions resulting from chemical mapping represent only a fraction of nucleosomes present in the cell, probably a small fraction.

Response: We appreciate the reviewer's comment. Our decision to pursue mammalian chemical mapping was informed by findings in fission yeast, where replacing 2 out of 3 H4 genes with H4S47C still produced chemical cleavage patterns comparable to those in strains with all three genes mutated. In H4S47C-expressing HeLa S3 cells (clones 1-2 and 2), RNA-seq analysis showed that 76.7 percent and 77.8 percent of total H4 mRNAs, respectively, originated from the RNAi-resistant H4S47C transgene. These results confirm that H4S47C is the predominant H4 transcript in these cells, directly addressing the reviewer's concern. This finding supports the suitability of our approach for generating genome-wide nucleosome maps, a point that is also relevant to the reviewer's subsequent comment.

Further, as the H4S47C gene is constitutively expressed from the CAG promoter that is NOT cell cycle-regulated, in contrast to endogenous histone H4 gene promoters, the H4S47C-containing nucleosomes do not represent the nucleosomes with canonical histones that are deposited in a replication-coupled way, but rather nucleosomes that contain variant histones, as, for example, also true for H3.3, that are deposited/exchanged in a replication-independent way. This profoundly changes the interpretation of the obtained nucleosome patterns. For example, take the chemical mapping interphase pattern of Cluster 6 in Fig. 3D. If the chemical mapping reflected the bulk of nucleosomes, then it would mean that the nucleosomes in the NDR (as detected by MNase-seq, Fig. S4B) and in the upstream region, have actually higher occupancy than the nucleosomes in the downstream coding region. A similar observation was already seen in mESCs (Fig. 3B in Voong et al., 2016, Cell. Also here the nucleosome occupancy in the NDR and upstream region relative to the TSS seemed higher than over the coding region. However, this is misleading. As the chemically mapped nucleosome positions monitor the constitutively expressed H4S47C in contrast to the replication-dependent wild-type H4, these high occupancies in the NDR and upstream region of Cluster 6 (or in mESCs) mainly reflect high histone turnover. (If such positions shall be called "fragile nucleosome" or else is a different point of debate.) It has to be realized that chemical mapping of nucleosomes with the authors' approach, in both the mouse and human cells, does not monitor the global majority of nucleosomes, as these still mainly contain wild-type H4, but a minority at most positions and a majority only at positions with high nucleosome turnover where canonical H4 is much less deposited due to low expression levels outside of S phase. This also explains why the authors detect chemically mapped nucleosomes over CTCF sites (Figs. 1D, 5C,D,E, S1G). This does not mean that CTCF cannot bind there or co-occupies its binding site with the histone octamer. It just means that TF binding sites have high turnover of TFs binding and unbinding and occasionally a nucleosome competing at the binding site. While this is an interesting aspect of the authors' chemical mapping approach, it has to be acknowledged. Otherwise their maps, which could be of great use, are misleading.

Response: The reviewer raises an important concern regarding the interpretation of chemically mapped nucleosomes, given that H4S47C is expressed from a constitutive CAG promoter and may preferentially label nucleosomes in regions with high histone turnover. In our design, we deliberately omitted a tag on H4S47C to minimize potential functional disruption, which unfortunately prevents us from distinguishing its protein-level distribution from that of endogenous H4.

To address this issue, we performed RNA-seq and found that H4S47C is the predominant H4 transcript in our cell lines. This finding suggests that H4S47C is broadly incorporated throughout the genome, not limited to regions of high turnover in transgene stably transfected cells. While we cannot entirely rule out incorporation bias, our chemical maps cover both euchromatic and heterochromatic regions, providing genome-wide representation.

Finally, our use of the term “fragile nucleosomes” is not intended to imply a specific structural configuration. Rather, it reflects chemically defined H4-associated nucleosome positions that may include high-turnover nucleosomes, hexasomes, or tetrasomes.

5) The authors repeatedly find that nucleosomes are more densely packed in interphase vs. metaphase cells (Figs. Figs. 2D,E, 3E,G 4G, 5B, S3) and correlate this with the higher transcription levels in interphase. It has been known since long (e.g., PMID 21602827) that nucleosome linker length decreases with increased transcription. The authors have to acknowledge such prior work.

Response: Yes, we appreciate this point and have cited the referenced work (PMID 21602827) in both the Results and Discussion sections to acknowledge prior findings on the relationship between transcription activity and nucleosome linker length.

6) In general, it seems that the authors could address more biologically interesting questions with their approach and data sets, e.g., the question of “bookmarking” during mitosis (see, for example, ref. 41).

Response: We agree that this is an interesting and important future direction. The question of transcriptional “bookmarking” during mitosis is highly relevant, and we believe that our data and approach could be valuable for addressing it in future studies.

Minor points:

Legend to Fig. S4B should refer to Fig. 3D not 4D.

Line 85: “In contrast, only 1.43% were found...” It should be made clearer what was “found”.

Response: We apologize for the error introduced during the rearrangement of figures. The revised manuscript and associated figure legends have been carefully updated with new data.

Referee #4

(Remarks to the Author)

In the manuscript titled 'Chemical mapping reveals differential nucleosome organization of human interphase and metaphase chromosomes,' the authors utilize their previously developed biochemical methods in “Insights into Nucleosome Organization in Mouse Embryonic Stem Cells through Chemical Mapping” published on Cell to detect the positioning of nucleosomes in HeLa cells during interphase and metaphase stages. They extensively compare the differential distribution of nucleosome positioning at gene loci, enhancers, CTCF binding sites, and other locations between these two phases. However, overall, the article predominantly focuses on describing observations and lacks necessary biological mechanisms. Therefore, I believe this manuscript is not suitable for publication in *[journal name redacted]*

Response: We respectfully disagree with the reviewer’s conclusion. While the study is primarily observational, we believe that this inductive analysis using chemical biology provides valuable new insights into genome organization across cell cycle phases. Our results offer a high-resolution resource and reveal dynamic changes in nucleosome positioning at key regulatory regions, laying the groundwork for future mechanistic investigations. We hope the reviewer agrees that the revised manuscript is now well-suited for publication in *Molecular Systems Biology* as a valuable resource.

The article only utilizes the HeLa cell line for chemical nucleosome mapping during interphase and metaphase. I believe that to elucidate differences in nucleosome positioning between interphase and metaphase, exploration in more cell lines is necessary; relying solely on one cell line is overly limiting.

Response: We agree with the reviewer that it is important to assess whether our findings are generalizable beyond a single cell line. In this study, we focused on generating the first human chemical nucleosome maps. To strengthen the conclusions, we have now validated our key findings using new chemical mapping data from an independent HeLa S3 cell line. These results are included in both the main text and Appendix of the revised manuscript.

The authors conducted two periods with two replicates each, but the results do not differentiate between these repetitions. In many figures, such as Fig. 2D, Fig. 3E, etc., it is unclear whether the differences between interphase and metaphase are due to experimental error or real results. I recommend presenting the results of each replicate separately to clarify this.

Response: Please also see our response to Reviewer 1's related comment. We agree that presenting individual replicate results is essential for evaluating experimental variability and reproducibility. In the revised manuscript, we generated separate chemical maps for each replicate and quantified variation between them (Fig. EV2A–C). We also included genome browser screenshots comparing individual and merged datasets from clone 2 (Fig. EV2D–F). Additionally, representative tracks from two independent clones (1-2 and 2) are shown in Fig. 1E–G, and further data from clone 1-2 are provided in the Appendix. Together, these analyses demonstrate that the observed differences between interphase and metaphase are robust and reproducible across replicates and clones.

While the concept of nucleosome center positioning score (NCP score) was mentioned in the article without detailed explanation, merely citing the reference, considering its frequent recurrence later in the text, it is suggested to provide a brief explanation in the article. This would help readers better understand and follow the content of the article.

Response: We appreciate the reviewer's suggestion. In the revised manuscript, we have added a brief explanation of the nucleosome center positioning (NCP) score in both the Results and Methods sections to aid reader understanding.

In Fig. 2D, Fig. 3E, Fig. 4G, and Fig. 5B, the authors respectively present frequency distribution plots of linker length (bp) near whole genome, TSS, enhancer, and CTCF binding sites during interphase and metaphase. They show similar trends across these figures. It is difficult to ascertain whether these trends represent differences across the whole genome or specifically at genomic loci such as TSS, enhancers, and CTCF sites.

Concerning Fig. 2E, it is argued that the conclusion about the differences in shorter linker lengths under 30 bp between E1-E4 and E9/E10/E11 compared to E5, E6, and E18 cannot be directly inferred from the figures, especially considering the high degree of similarity in the distribution curves between E9/E10/E11 and E5, E6.

Response: We thank the reviewer for the comment. In this manuscript, we used probability density plots to visualize linker length distributions in interphase and metaphase across selected

chromatin states and genomic landmarks such as TSSs, enhancers, and CTCF binding sites. To address the reviewer's concern, we have now included a Wald test analysis to assess whether the enrichment of short linker lengths in interphase is significantly greater in specific regions compared to other regions or the rest of the genome. These statistical results are provided in Appendix Tables S2 and S3. Briefly, the enrichment of shorter linker lengths reported in the original manuscript remains highly significant, with p-values equal to 0.

Moreover, the article does not provide clear conclusions derived from comparisons in this section. It is unclear what the article aims to draw from comparing different chromatin states in the context of shorter linker lengths under 30 bp.

Response: The conclusion is that short linker lengths (≤ 30 bp) are significantly more enriched in interphase compared to metaphase, likely due to increased transcriptional activity. As noted by Reviewer 3, previous studies have similarly shown that internucleosome spacing decreases as transcription increases (PMID 21602827). Our analysis extends this observation beyond transcription start sites (TSSs) to additional regulatory landmarks, including enhancers and CTCF binding sites. We find that nucleosome arrays become denser near these cis-regulatory elements during the transition from metaphase to interphase.

These observations lack statistical significance. The absence of significance markers in figures suggests that if the authors aim to demonstrate differences in nucleosome linker length at specific locations, robust hypothesis testing is necessary. The presentation of results in the manuscript lacks persuasiveness.

Response: We thank the reviewer for this important comment. As noted above, we have now conducted rigorous hypothesis testing using the Wald test, with results presented in Appendix Tables S2 and S3.

First, we tested whether the enrichment of short linker lengths (interphase vs. metaphase) is significantly greater in E1-E4 or E9-E11 states compared to E5-E6, E13, or E18 (Appendix Table S2). Additionally, we tested whether the enrichment of short linker lengths is significantly greater in defined TSS regions, enhancer regions, or CTCF binding sites compared to non-TSS, non-enhancer, or non-CTCF regions (Table S3).

All tests showed extreme statistical significance, with p-values virtually equal to 0.

In the "Nucleosome repositioning at TSS during mitosis" section, it is recommended to provide further explanation of the "-1 nucleosome peak" concept.

Response: An excellent review (PMCID: PMC10168609) provides a comprehensive overview of the "-1 nucleosome" concept. To maintain brevity, we cited this reference and included a brief description in the manuscript.

In Fig. 4C and D, the authors classify enhancers into 8 clusters based on nucleosome occupancy patterns near enhancers. However, there is no further discussion regarding these 8 clusters in the manuscript. It is interesting to explore which genes these enhancers correspond to, whether these genes exhibit differential expression between the two phases, and if they are associated with phase transitions. I recommend that the authors delve deeper into these biological questions rather than solely observing the phenomena.

Response: We thank the reviewer for the suggestion. The genomic regions assigned to each cluster are scattered throughout the genome. We did not observe any meaningful enrichment associated with individual clusters.

In figures 4E and 4F, the article only selected two enhancer sites from cluster 2 and cluster 7. However, as stated in the article, cluster 1-4 and 7, 8 all showed a significant decrease in local correlation from ~0.6 to 0.1 or 0.2. The reason for selecting only enhancer sites from cluster 2 and 7 is not explained. Additionally, the article mentioned that cluster 5 and 6 showed a peak at the predicted enhancer sites but with minor drops at the sides, which warrants further investigation.

Response: Due to space constraints in Figure 4 and the limited number of Expanded Figures permitted by the journal, we were only able to present a subset of representative examples. We selected clusters 2 and 7 because they display distinct and contrasting correlation patterns, effectively demonstrating that our clustering analysis captures differential nucleosome organization near enhancers. Additional examples from other clusters would restate the same conclusion without providing substantially new insights.

In the "Nucleosome positioning is preserved at exon/intron boundaries during mitosis" section, it is suggested to further elaborate on the biological significance of NET-seq read scores and their relationship with NCP scores.

Response: It has been proposed that transcription kinetics influences mRNA splicing outcomes (PMC3038581, PMCID: PMC4304646). NET-seq and GRO-seq studies have shown the accumulation of paused RNA polymerase II at exon–intron junctions. Nucleosomes positioned at these junctions can act as physical barriers, promoting RNAPII pausing and facilitating the assembly of the splicing machinery. We previously described this mechanism and its potential significance in the mouse ES cell chemical map. In the current study, we focused on a new finding: nucleosomes at these exon–intron junctions remain well-positioned during mitosis, even in the absence of active transcription, a phenomenon not previously reported.

Finally, considering Fig. 6B is a partly zoomed-in version of Fig. 6A, combining Fig. 6A and 6B into one figure could enhance readability for the readers.

Response: We thank the reviewer for the helpful suggestion. To improve clarity when referring to individual panels in the text, we have chosen to keep them as separate panels, now labeled Fig. 7A and 7B in the revised Figure 7.

11th Sep 2025

Manuscript Number: MSB-2025-13024-T

Title: Differential nucleosome organization in human interphase and metaphase chromosomes

Dear Prof Wang,

Thank you again for submitting your revised work to Molecular Systems Biology. We have now heard back from one of the original reviewers (Reviewer #3) as well as two new reviewers (Reviewers #5 and #6) who we asked to evaluate your study given the previous reviewers' comments and your responses. As you will see below, the reviewers are globally supportive on the value of your work and novelty of the dataset. However, Reviewer #3 continues to point out that some of their comments were not fully addressed, potentially due to ambiguous wording. Reviewer #5 also suggests some minor additional analyses and Reviewer #6 has a few suggestions on presentation of data and a discussion point. We would therefore ask you to address their concerns in a revision. Please let me know in case you would like to discuss in further detail any of the any of the reviewer comments, I would be happy to schedule a call.

When submitting your revised manuscript, please carefully review the instructions that follow below. We require:

1) A .docx formatted version of the manuscript text (including legends for main figures, EV figures and tables). Please make sure that the changes are highlighted to be clearly visible. Alternatively you may choose to submit your manuscript as a LaTeX file.

4) A .docx formatted letter INCLUDING the reviewers' reports and your detailed point-by-point responses to their comments. As part of the EMBO Press transparent editorial process, the point-by-point response is part of the Peer Review File (PRF), which will be published alongside your paper.

5) A complete author checklist, which you can download from our author guidelines (<https://www.embopress.org/page/journal/17574684/authorguide#submissionofrevisions>). Please insert information in the checklist that is also reflected in the manuscript. The completed author checklist will also be part of the PRF.

6) Please note that all corresponding authors are required to supply an ORCID ID for their name upon submission of a revised manuscript.

7) It is mandatory to include a 'Data Availability' section after the Materials and Methods. Before submitting your revision, primary datasets produced in this study need to be deposited in an appropriate public database, and the accession numbers and database listed under 'Data Availability'. Please remember to provide a reviewer password if the datasets are not yet public (see <https://www.embopress.org/page/journal/17574684/authorguide#dataavailability>).

In case you have no data that requires deposition in a public database, please state so in this section as follows: "This study includes no data deposited in external repositories". Note that the Data Availability Section is restricted to new primary data that are part of this study.

8) All Materials and Methods need to be described in the main text using our 'Structured Methods' format, which is required for all research articles. According to this format, the Methods section includes a Reagents and Tools Table (listing key reagents, experimental models, software and relevant equipment and including their sources and relevant identifiers) followed by a Methods and Protocols section describing the methods using a step-by-step protocol format. The aim is to facilitate adoption of the methodologies across labs. Please upload the Reagents and Tools table as a separate document when submitting your revised manuscript. More information on how to adhere to this format as well as a downloadable template (.docx) for the Reagents and Tools Table can be found in our author guidelines: <https://www.embopress.org/page/journal/17444292/authorguide#structuredmethods>

An example of a Method paper with Structured Methods can be found here: <https://www.embopress.org/doi/10.15252/msb.20178071>.

9) For data quantification: please specify the name of the statistical test used to generate error bars and p-values, the number

(n) of independent experiments (specify technical or biological replicates) underlying each data point and the test used to calculate p-values in each figure legend. The figure legends should contain a basic description of n, p-values and the test applied. Graphs must include a description of the bars and the error bars (s.d., s.e.m.). Please provide exact p-values (in either the figure or figure legend).

10) Our journal encourages inclusion of *data citations in the reference list* to directly cite datasets that were re-used and obtained from public databases. Data citations in the article text are distinct from normal bibliographical citations and should directly link to the database records from which the data can be accessed. In the main text, data citations are formatted as follows: "Data ref: Smith et al, 2001" or "Data ref: NCBI Sequence Read Archive PRJNA342805, 2017". In the Reference list, data citations must be labeled with "[DATASET]". A data reference must provide the database name, accession number/identifiers and a resolvable link to the landing page from which the data can be accessed at the end of the reference. Further instructions are available at .

11) We replaced Supplementary Information with Expanded View (EV) Figures and Tables that are collapsible/expandable online. EV Figures should be cited as 'Figure EV1, Figure EV2' etc... in the text and their respective legends should be included in the main text after the legends of regular figures.

- Additional Tables/Datasets should be labeled and referred to as Table EV1, Dataset EV1, etc. Legends should be provided in a separate tab in case of .xls files. Alternatively, the legend can be supplied as a separate text file (README) and zipped together with the Table/Dataset file.

<https://www.embopress.org/page/journal/17574684/authorguide#expandedview>

12) Author contributions: CRedit has replaced the traditional author contributions section because it offers a systematic machine-readable author contributions format that allows for more effective research assessment. Please remove the Authors Contributions from the manuscript and use the free text boxes beneath each contributing author's name in our system to add specific details on the author's contribution. More information is available in our guide to authors.

13) Disclosure statement and competing interests: We updated our journal's competing interests policy in January 2022 and request authors to consider both actual and perceived competing interests. Please review the policy <https://www.embopress.org/competing-interests> and update your competing interests if necessary.

14) Every published paper now includes a 'Synopsis' to further enhance discoverability. Synopses are displayed on the journal webpage and are freely accessible to all readers. They include a short stand first (maximum of 300 characters, including space) as well as 2-5 one-sentences bullet points that summarizes the paper. Please write the bullet points to summarize the key NEW findings. They should be designed to be complementary to the abstract - i.e. not repeat the same text. We encourage inclusion of key acronyms and quantitative information (maximum of 30 words / bullet point). Please use the passive voice. Please attach these in a separate file or send them by email, we will incorporate them accordingly.

Please note that these would be the final versions and changes during proofing are usually not allowed.

15) As part of the EMBO Publications transparent editorial process initiative (see our policy here:

https://www.embopress.org/transparent-process#Review_Process), Molecular Systems Biology will publish online a Peer Review File (PRF) to accompany accepted manuscripts.

In the event of acceptance, this file will be published in conjunction with your paper and will include the anonymous referee reports, your point-by-point response and all pertinent correspondence relating to the manuscript. Let us know whether you agree with the publication of the PRF and as here, if you want to remove or not any figures from it prior to publication.

Please note that the Author checklist will be published at the end of the PRF.

Molecular Systems Biology has a "scooping protection" policy, whereby similar findings that are published by others during review or revision are not a criterion for rejection. Should you decide to submit a revised version, I do ask that you get in touch after three months if you have not completed it, to update us on the status.

Yours sincerely,

Poonam Bheda

Poonam Bheda, PhD
Scientific Editor

Reviewer #3:

The authors addressed my major point no. 1 regarding adding a second independent PiggyBac clone (clone 2). While comparison of their chemical mapping data at three genome loci between clone 1-2 and 2 in Fig. 1D-F looks ok, it represents only a tiny fraction of the genome-wide data. To allow the reader to better evaluate the biological reproducibility of this chemical mapping approach, the authors have to employ and show in their study the same rigorous statistical analyses for the biological replicates (comparison between clones) as they did for the comparison of technical replicates (between different sequencing runs) as shown in Fig. EV2A-C. (Note that the legend to Fig. EV2A erroneously states "biological replicates" even though this is a comparison between technical replicates within data for clone 2 only, at least as explained in the authors' rebuttal.) My guess is that the comparisons in Fig. EV2A-C look so good as they consider only technical sequencing replicates, while the true biological variation will be considerably larger. This need not be a problem for this study, but it should be presented nonetheless. If done, which should not pose any problems, I consider this point settled.

The authors' reply to my major point no. 4 is not sufficient. It is not clear if the results presented in Fig. EV1B-D compare the HeLaS3 parental cells with the clones 1-2 and 2 under conditions before or after siRNA-mediated ablation of H4 expression. While both comparisons are interesting (also commented by Reviewer #1), the latter one is the more important comparison as it pertains to the conditions of the actual nucleosome mapping. The authors have to clarify. The high percentage (around 77%) of mutant H4 mRNA expression may argue for conditions after siRNA treatment, while the undisturbed growth curves in Fig. EV1C argues for conditions prior to siRNA treatment. I still find it hard to believe that cell growth would not be compromised if ablation of WT H4 is effective and mainly a non-cell cycle-regulated mutant H4 allele is expressed.

I appreciate that it is difficult to distinguish the WT from the mutated H4 without using tags, which may introduce complications on their own. But at least on the level of growth curves and mRNA expression, the effects after siRNA treatment should be reported. I do not expect that everything looks like WT after siRNA treatment and this would not compromise the value of this study. Nonetheless, the reader needs to know.

Similarly, the authors' reply to my comments (part of my major point 4) regarding the correlation of chemical mapping via the mutant H4 variant and histone exchange is evasive. Even if the mutant H4 is predominantly expressed and incorporated throughout the genome, this does not address the incorporation bias issue. While the authors cannot do much about it, this point should be discussed in the manuscript and not just in the rebuttal.

Collectively, I do think that this is a great study and should be published and will be a valuable resource and the methodological approach is promising in many ways. Nonetheless, I do demand that this is properly reported (genome-wide variation between biological replicates, relative levels of WT vs. mutant H4 AFTER siRNA treatment, if not shown already, e.g., by mRNA levels, effect on growth rate AFTER siRNA treatment) and properly discussed (include comment on incorporation bias due to non-cell cycle regulation of mutant H4).

All other points I raised previously are sufficiently addressed.

Reviewer #5:

As seen in the manuscript and responses to reviews, the authors improved the manuscript substantially. Particularly, they supplemented adequate experiments and demonstrated the robustness of the chemical-mapping approach for inferring nucleosome positions. Based on the approach, the first chemical map of nucleosomes for human genome was obtained, and some interesting findings are also presented. I only have several minor comments for the authors.

1. In the section "Nucleosome repositioning at TSS during mitosis", it was shown that strong -1 nucleosomes detected by the chemical method in Clusters 2 and 6 were absent in the MNase map, likely due to their fragility. Is it possible to evaluate this by analyzing reads of ~80 bp in the MNase-seq experiment? In other words, if it was due to nucleosome fragility, nucleosome peak signals could be expected when reads of ~80 bp were analyzed.

2. When analyzing the difference in nucleosome positioning between interphase and metaphase, the authors employed a correlation-based approach. Why not using a more direct method to measure the difference? For example, Fold-Change and differential peaks can be obtained along with p-value by using statistical tests as described in Chen et al., DANPOS: dynamic analysis of nucleosome position and occupancy by sequencing. *Genome Res.* 2013. This is not mandatory if the authors can convince me that their results based on correlation analysis are solid.

3. It is unclear where the representative active enhancers are in Figure 4 E-F.

4. Could the authors give some discussion about the possible coupling of nucleosome dynamics during mitosis (e.g. difference between interphase and metaphase) with TAD dynamics during mitosis? For example, how does nucleosome occupancy alter at the boundaries of disappeared or newly occurred TADs during mitosis? This might also be associated with the results regarding CTCF binding sites, as they are primary source of loop anchor sites and TAD boundaries.

5. In the manuscript, they stated that DNA's intrinsic bendability plays a significant role in determining nucleosome dyad positioning. In this regard, a deformation energy-based model might be useful for their further study (Liu et al. A deformation energy model reveals sequence-dependent property of nucleosome positioning. *Chromosoma*, 2021).

6. As described in the manuscript, high nucleosome density is a possible feature of high transcription activity. But in some literatures, it was proposed that lower dip of NDR near TSS is correlated with high transcription level, and high nucleosome density or occupancy at the NDR-downstream regions might inhibit RNA pol II progression. What is your opinion?

Reviewer #6:

The work by Li et al., addresses an important question in chromatin biology: how mitotic chromosome condensation affects nucleosome positioning and whether nucleosome organization is mostly preserved or actively rearranged between cell-cycle stages. The authors provide convincing evidence for localized nucleosome repositioning at key regulatory elements during interphase, while reinforcing that broad swathes of the genome (especially gene bodies and heterochromatin) remain largely unchanged. The central conclusions, that nucleosome organization is broadly similar overall, with specific, functionally important exceptions, are supported by the analyses, though the phrasing of "distinct organizational principles" might be somewhat overstated. The authors' own cumulative distribution analysis indicates that only a minority of regions (on the order of a few percent of the genome) undergo dramatic repositioning. Thus, the evidence suggests nuanced, site-specific differences rather than a wholesale genome-wide reorganization.

Technically the work appears robust especially after the addition of new clones and replicates. I would like to see the raw dyad midpoints (as mapped by the chemical cleavage) overlaid on the smoothed nucleosome profiles (plot the dyad positions relative to the X-axis and use jitter to spread the points out on the y axis). This will address my primary potential concern of sparseness and overinterpretation of the smoothed profiles.

It would also be worthwhile to mention in the discussion expected differences in nucleosome positioning within interphase. For example, nucleosomes deposited by DNA replication may differ from those remodeled by transcription.

Point-by-point response (Revision 2)

Reviewer #3:

The authors addressed my major point no. 1 regarding adding a second independent PiggyBac clone (clone 2). While comparison of their chemical mapping data at three genome loci between clone 1-2 and 2 in Fig. 1D-F looks ok, it represents only a tiny fraction of the genome-wide data. To allow the reader to better evaluate the biological reproducibility of this chemical mapping approach, the authors have to employ and show in their study the same rigorous statistical analyses for the biological replicates (comparison between clones) as they did for the comparison of technical replicates (between different sequencing runs) as shown in Fig. EV2A-C. (Note that the legend to Fig. EV2A erroneously states "biological replicates" even though this is a comparison between technical replicates within data for clone 2 only, at least as explained in the authors' rebuttal.) My guess is that the comparisons in Fig. EV2A-C look so good as they consider only technical sequencing replicates, while the true biological variation will be considerably larger. This need not be a problem for this study, but it should be presented nonetheless. If done, which should not pose any problems, I consider this point settled.

Response: We thank the reviewer for this helpful comment. We would like to clarify the distinction between technical and biological duplicates in our analyses. In the initial review, we understood the reviewer's suggestion as referring to technical duplicates, meaning different sequencing runs of the same library. During the revision, however, we generated completely new datasets from an independently derived chemical mapping cell line (clone 2). To address the concern about reproducibility using a more stringent criterion, the duplicate data presented were obtained from two independent chemical mapping experiments performed on the same cell line (clone 2), which we consider biological duplicates.

Figure S3
To directly compare across clones, we have now included the requested genome wide comparison between clone 1 and clone 2 (Fig. EV2G and Fig. S3). As shown in Fig. S3, data from the two clones are well correlated, although as expected the correlations are lower than those observed between duplicates of the same clone (see Fig. EV2A-B). We note that these datasets were collected three years apart, by different researchers, and sequenced on different Illumina platforms.

Regarding the examples in Fig. 1, they were selected in the first submission before the generation of clone 2 maps.

The authors' reply to my major point no. 4 is not sufficient. It is not clear if the results presented in Fig. EV1B-D compare the HeLaS3 parental cells with the clones 1-2 and 2 under conditions before or after siRNA-mediated ablation of H4 expression. While both comparisons are interesting (also commented by Reviewer #1), the latter one is the more important comparison as it pertains to the conditions of the actual nucleosome mapping. The authors have to clarify. The high percentage (around 77%) of mutant H4 mRNA expression may argue for conditions after siRNA treatment, while the undisturbed growth curves in Fig. EV1C argues for conditions prior to siRNA treatment. I still find it hard to believe that cell growth would not be compromised if ablation of WT H4 is effective and mainly a non-cell cycle-regulated mutant H4 allele is expressed.

Response: We thank the reviewer for raising this important question. We would like to clarify the process by which we generated mammalian cells suitable for chemical mapping. The key step was to introduce both the H4-shRNA construct and the RNAi-resistant H4 transgene simultaneously, thereby preventing depletion of total H4 levels through efficient integration of multiple PiggyBac transposons. From our experience, this approach consistently produces surviving clones that are resistant to both puromycin and hygromycin selections linked to the shRNA and H4 transgenes. As illustrated in the accompanying diagram, there was no distinct “before” or “after” siRNA stage during the generation of chemical mapping cell lines.

To directly address the reviewer’s question, the high percentage of mutant H4 mRNA corresponds to the state “after siRNA” the reviewer referred to. Regarding cell growth, our experimental procedure inherently selects for clones with normal proliferation. Cells that exhibit strong growth defects after transfection do not survive the dual selection and therefore are not analyzed further for chemical mapping suitability. In other words, we intentionally selected clones that maintained normal growth and morphology characteristics. Although cell cycle-regulated wild-type H4 expression was reduced or absent in these cells, this reduction did not produce a measurable effect on overall cell growth of the chosen chemical mapping cell lines.

I appreciate that it is difficult to distinguish the WT from the mutated H4 without using tags, which may introduce complications on their own. But at least on the level of growth curves and mRNA expression, the effects after siRNA treatment should be reported. I do not expect that everything looks like WT after siRNA treatment and this would not compromise the value of this study. Nonetheless, the reader needs to know.

Response: As outlined above, there was no distinct “before siRNA” stage to compare with an “after siRNA” condition, because the H4-shRNA construct and the RNAi-resistant H4 transgene were co-

introduced during the establishment of chemical mapping cell lines. Clones were preselected based on normal growth and stable H4 expression. Accordingly, the data presented reflect the post-siRNA condition, under which cells exhibited normal proliferation despite the reduction of wild-type H4.

Similarly, the authors' reply to my comments (part of my major point 4) regarding the correlation of chemical mapping via the mutant H4 variant and histone exchange is evasive. Even if the mutant H4 is predominantly expressed and incorporated throughout the genome, this does not address the incorporation bias issue. While the authors cannot do much about it, this point should be discussed in the manuscript and not just in the rebuttal.

Response: We agree with the reviewer that the mutant H4 is not cell cycle regulated and may exhibit incorporation bias. We have added this consideration to the main text to clarify our interpretation. It is likely that continuous expression and incorporation of the mutant H4 over multiple cell divisions promotes gradual replacement of preexisting histones through replication-coupled and replication-independent exchange mechanisms. Over extended culture periods, this process would help equilibrate mutant H4 incorporation across the genome in chemical mapping analyses.

Collectively, I do think that this is a great study and should be published and will be a valuable resource and the methodological approach is promising in many ways. Nonetheless, I do demand that this is properly reported (genome-wide variation between biological replicates, relative levels of WT vs. mutant H4 AFTER siRNA treatment, if not shown already, e.g., by mRNA levels, effect on growth rate AFTER siRNA treatment) and properly discussed (include comment on incorporation bias due to non-cell cycle regulation of mutant H4). All other points I raised previously are sufficiently addressed.

Response: We thank the reviewer for the thoughtful suggestions and positive assessment of our study. We have addressed the reviewer's concerns regarding variation among biological replicates, H4 mRNA expression levels, and cell growth phenotypes (see above). In addition, we have included a paragraph in the revised manuscript discussing the potential incorporation bias that may arise from the cell cycle-independent expression of the mutant H4.

Reviewer #5:

As seen in the manuscript and responses to reviews, the authors improved the manuscript substantially. Particularly, they supplemented adequate experiments and demonstrated the robustness of the chemical-mapping approach for inferring nucleosome positions. Based on the approach, the first chemical map of nucleosomes for human genome was obtained, and some interesting findings are also presented. I only have several minor comments for the authors.

1. In the section "Nucleosome repositioning at TSS during mitosis", it was shown that strong -1 nucleosomes detected by the chemical method in Clusters 2 and 6 were absent in the MNase map, likely due to their fragility. Is it possible to evaluate this by analyzing reads of ~80 bp in the MNase-seq experiment? In other words, if it was due to nucleosome fragility, nucleosome peak signals could be expected when reads of ~80 bp were analyzed.

Response: In our previous study (PMC5135608), we performed a similar analysis to examine the relationship between fragile nucleosome and ~80 bp MNase fragments. Unfortunately, in the current study, our MNase-seq data were generated using mono-nucleosome fragments (~150 bp) from the chemical mapping cell lines. Therefore, we do not have corresponding short-fragment data to conduct this analysis at present.

2. When analyzing the difference in nucleosome positioning between interphase and metaphase, the authors employed a correlation-based approach. Why not using a more direct method to measure the difference? For example, Fold-Change and differential peaks can be obtained along with p-value by

using statistical tests as described in Chen et al., DANPOS: dynamic analysis of nucleosome position and occupancy by sequencing. *Genome Res.* 2013. This is not mandatory if the authors can convince me that their results based on correlation analysis are solid.

Response:

We thank the reviewer for this suggestion. There are several caveats in directly applying the methods described in the DANPOS paper to our dataset. First, our analysis focuses on global or regional differences in nucleosome positioning patterns rather than changes at single base-pair resolution. Compared with the MNase-based maps used in DANPOS, our chemical mapping data already provide base-pair resolution of nucleosome dyads. If desired, we could have calculated the fold change or positional difference between NCP scores at each nucleotide; however, this was not the main goal of our study. Our objective was to assess broader-scale pattern changes, for which local correlation analysis provides a simple yet effective metric.

In addition, there are methodological and statistical limitations in adopting the DANPOS approach for our chemical mapping data. As acknowledged by the authors of DANPOS, “neither fold change nor direct subtraction appears to be a good measurement for dynamic nucleosomes.” Fold change is not statistically appropriate for hypothesis testing because it does not account for variance. Furthermore, DANPOS requires extremely deep sequencing coverage (on the order of >200-fold) to reliably detect dynamic positioning, whereas our chemical mapping data have much lower coverage.

DANPOS also tests read counts at every individual base pair, which presents additional complications when applied to human chemical mapping data. Owing to the large genome size in mammals, the cleavage counts in our chemical maps are typically low, often in single digits or zero. The Poisson and Chi-square tests used by DANPOS are unreliable in this context: the Poisson test assumes equality of mean and variance (which is typically violated due to overdispersion in high throughput sequencing data), and the Chi-square test requires asymptotic normality, which is invalid when counts are sparse (as in a 2×2 table comparing a single base pair to the rest of the genome). Even using a negative binomial model would be challenging, since overdispersion varies across genomic loci and becomes difficult to model accurately when counts are extremely low. We are concerned that applying these tests to the data in this study may result in false results and subsequent misleading interpretations.

Given these considerations, we believe that local correlation analysis provides a more reliable and interpretable approach to compare nucleosome positioning patterns between interphase and metaphase in our study.

3. It is unclear where the representative active enhancers are in Figure 4 E-F.

Response: We have now included the precise genomic coordinates of the two representative active enhancers in Figure 4E and 4F.

4. Could the authors give some discussion about the possible coupling of nucleosome dynamics during mitosis (e.g. difference between interphase and metaphase) with TAD dynamics during mitosis? For example, how does nucleosome occupancy alter at the boundaries of disappeared or newly occurred TADs during mitosis? This might also be associated with the results regarding CTCF binding sites, as they are primary source of loop anchor sites and TAD boundaries.

Response: We thank the reviewer for raising this important question. There are, however, technical limitations that currently prevent us from obtaining informative results to directly address this point. While the high resolution of chemical mapping enabled us to reveal new features of nucleosome reorganization during mitosis, these features can only be fully interpreted in the context of high-resolution genomic landmarks. The currently available Hi-C datasets have limited resolution due to both the cleavage-based mapping method and sequencing depth. For example, the resolution of the

available HeLa S3 Hi-C dataset is approximately 50 kb (<https://cb.csail.mit.edu/tadmap/>). Therefore, direct analysis of nucleosome features at TAD boundaries is not yet informative at this scale.

In response to the reviewer's suggestion, we identified a subset of CTCF binding sites located within a ± 25 kb window of reported TAD boundaries and analyzed nucleosome occupancy around these sites, as shown in the figure below. The analysis revealed that nucleosome occupancy scores at these CTCF sites were higher in top three quantiles (CTCF ChIP-seq signals) compared with the genome average (main figure Fig. 5D). Thus, at least when aligned to CTCF sites, the nucleosome organization patterns we described appear to be conserved at TAD boundaries.

5. In the manuscript, they stated that DNA's intrinsic bendability plays a significant role in determining nucleosome dyad positioning. In this regard, a deformation energy-based model might be useful for their further study (Liu et al. A deformation energy model reveals sequence-dependent property of nucleosome positioning. *Chromosoma*, 2021).

Response: We agree that deformation energy represents an important factor influencing DNA cyclizability and nucleosome positioning. However, it is not the only determinant. Our current analysis focuses on chemical mapping data and C-scores that are either experimentally measured or derived directly from such data. We believe that future modeling studies incorporating deformation energy parameters will be valuable for quantitatively interpreting and extending our experimental observations.

6. As described in the manuscript, high nucleosome density is a possible feature of high transcription activity. But in some literatures, it was proposed that lower dip of NDR near TSS is correlated with high transcription level, and high nucleosome density or occupancy at the NDR-downstream regions might inhibit RNA pol II progression. What is your opinion?

Response: We agree with the reviewer that elevated nucleosome occupancy downstream of the nucleosome-depleted region (e.g., at the +1 and +2 positions) is positively correlated with RNA polymerase II (RNAPII) pausing. This interpretation is consistent with our understanding and supported by our previous chemical mapping analyses (PMC5135608). Using the interphase maps generated in this study, we observed similar patterns (see below). For this analysis, the *pausing index* (PI) was defined as the ratio of average NET-seq density in the promoter region (TSS ± 150 bp) to the average GRO-seq density across the gene body (+250 to +2250 bp), based on data from Mayer *et al.* (2015;

PMC4528962). We then plotted nucleosome occupancy around the TSS in quartiles of PI scores (see figure), revealing that nucleosomes downstream of the TSS likely contribute to the regulation of RNAPII pausing. In the current manuscript, we present related data in Fig. 3C, which demonstrate a positive correlation between nucleosome occupancy and transcriptional activity. We note that RNAPII pausing represents a regulatory checkpoint during transcription elongation and is neither directly proportional to nor incompatible with transcriptional activity. Because this work primarily focuses on comparing interphase and metaphase nucleosome positioning, we did not include these additional analyses in the main figures due to space constraints.

Reviewer #6:

The work by Li et al., addresses an important question in chromatin biology: how mitotic chromosome condensation affects nucleosome positioning and whether nucleosome organization is mostly preserved or actively rearranged between cell-cycle stages. The authors provide convincing evidence for localized nucleosome repositioning at key regulatory elements during interphase, while reinforcing that broad swathes of the genome (especially gene bodies and heterochromatin) remain largely unchanged. The central conclusions, that nucleosome organization is broadly similar overall, with specific, functionally important exceptions, are supported by the analyses, though the phrasing of "distinct organizational principles" might be somewhat overstated. The authors' own cumulative distribution analysis indicates that only a minority of regions (on the order of a few percent of the genome) undergo dramatic repositioning. Thus, the evidence suggests nuanced, site-specific differences rather than a wholesale genome-wide reorganization.

Technically the work appears robust especially after the addition of new clones and replicates. I would like to see the raw dyad midpoints (as mapped by the chemical cleavage) overlaid on the smoothed nucleosome profiles (plot the dyad positions relative to the X-axis and use jitter to spread the points out on the y axis). This will address my primary potential concern of sparseness and overinterpretation of the smoothed profiles.

Response: We have changed the wording "distinct organizational principles" to "distinct patterns of nucleosome organization" in the abstract. We have performed the requested analysis. Both NCP scores and center-weighted nucleosome occupancies are now plotted separately at the three representative examples corresponding to Fig. 1D-F (see below). However, we were unable to include these additional tracks in the main figure due to space constraints.

It would also be worthwhile to mention in the discussion expected differences in nucleosome positioning within interphase. For example, nucleosomes deposited by DNA replication may differ from those remodeled by transcription.

Response: We thank the reviewer for raising this important point. Because the chemical mapping was performed on an asynchronous cell population, loci undergoing active DNA replication are likely underrepresented and may be masked by the more abundant G1-phase cells that are transcriptionally active. Nevertheless, this is a valid concern. We have now noted in the Discussion that the interphase nucleosome map likely represents a heterogeneous mixture of nucleosome positions, predominantly reflecting the G1 phase but also influenced by other stages of the cell cycle.

9th Dec 2025

Manuscript Number: MSB-2025-13024R

Title: Differential nucleosome organization in human interphase and metaphase chromosomes

Dear Prof Wang,

Thank you for the submission of your revised manuscript to Molecular Systems Biology. I am pleased to inform you that we will be able to accept your manuscript pending the following final amendments and appropriate response to reviewers:

1) In the main manuscript file, please include keywords to max. 5.

2) Please rename "Data and availability" to "Data availability". Please also remove the reviewer access information, publicly release the datasets, and format the 'Data availability' section according to the example below.

"The datasets and computer code produced in this study are available in the following databases:

- Chip-Seq data: Gene Expression Omnibus GSE46748 (<https://www.ncbi.nlm.nih.gov/geo/query/acc.cgi?acc=GSE46748>)

- Modeling computer scripts: GitHub (<https://github.com/SysBioChalmers/GECKO/releases/tag/v1.0>)

- [data type]: [full name of the resource] [accession number/identifier] ([doi or URL or identifiers.org/DATABASE:ACCESSION])"

3) Please include a README file on Github with practical use instructions for potential future users of your code.

4) Please rename "Competing interests" to "Disclosure and competing interests statement". We updated our journal's competing interests policy in January 2022 and request authors to consider both actual and perceived competing interests. Please review the policy <https://www.embopress.org/competing-interests> and update your competing interests if necessary.

5) Author contributions: Please remove it from the manuscript and specify author contributions in our submission system.

CRedit has replaced the traditional author contributions section because it offers a systematic machine-readable author contributions format that allows for more effective research assessment. You are encouraged to use the free text boxes beneath each contributing author's name to add specific details on the author's contribution. More information is available in our guide to authors:

<https://www.embopress.org/page/journal/17574684/authorguide#authorshipguidelines>

6) Our journal encourages inclusion of *data citations in the reference list* to directly cite datasets that were re-used and obtained from public databases. Data citations in the article text are distinct from normal bibliographical citations and should directly link to the database records from which the data can be accessed. In the main text, data citations are formatted as follows: "Data ref: Smith et al, 2001" or "Data ref: NCBI Sequence Read Archive PRJNA342805, 2017". In the Reference list, data citations must be labeled with "[DATASET]". A data reference must provide the database name, accession number/identifiers and a resolvable link to the landing page from which the data can be accessed at the end of the reference. Further instructions are available at .

7) In the Methods, please take care of the following:

- Cell lines: Please be sure to include a sentence in the Methods as to whether or not the cell lines were recently authenticated and tested for mycoplasma contamination. Please also be sure to update the Author Checklist with this information and where it can be found in the manuscript.

- Please ensure that a statement on whether or not blinding was done is included in the Methods even if no blinding was done. Please also be sure to update the Author Checklist with this information and where it can be found in the manuscript.

8) Please place individual sections of the manuscript in the following order with the correct nomenclature: Title page - Abstract & Keywords - Introduction - Results - Discussion - Methods - Data Availability - Acknowledgements - Disclosure and Competing Interests Statement - References - Figure Legends - Expanded View Figure Legends.

9) For the figures and figure legends, please take care of the following:

- Please make sure to update the callouts of all figures in the main manuscript text. Callouts are missing for Fig. 7E and Appendix Figures S4-S7. Callouts are also missing for the individual panels of Appendix figures. The callout for Fig. S3 should be corrected to Appendix Fig. S3.

10) Please remove the "Appendix information" section from the main manuscript.

11) In the Appendix PDF, please remove the "Appendix Supplementary Figure Legend" section and instead place the individual legends below each of the corresponding figures.

12) Synopsis:

- Synopsis text: Please provide a separate word document including a short standfirst (maximum of 300 characters, including spaces) and up to 5 bullet points to summarise the key NEW findings. They should be designed to be complementary to the abstract - i.e. not repeat the same text. We encourage inclusion of key acronyms and quantitative information (maximum of 30 words / bullet point). Please use the passive voice.

13) Please explain the Source Data for Figure 2B, as it is unclear how this matches the pie chart in the manuscript. In addition, the numbers are exactly the same in the columns for 'y', 'y min', and 'y max' - please explain what these numbers represent. Please also explain the Source Data for Figure 3G, as it is also unclear what numbers are represented in the bar graph.

14) As part of the EMBO Publications transparent editorial process initiative (see our policy here:

https://www.embopress.org/transparent-process#Review_Process), Molecular Systems Biology will publish online a Peer Review File (PRF) to accompany accepted manuscripts. This file will be published in conjunction with your paper and will include the anonymous referee reports, your point-by-point response and all pertinent correspondence relating to the manuscript. Let us know whether you agree with the publication of the PRF and as here, if you want to remove or not any figures from it prior to publication. Please note that the Authors checklist will be published at the end of the PRF.

15) After your paper is published, we may promote it on social media. If you have any handles or hashtags for Bluesky you would like included, please let us know.

16) Please provide a point-by-point letter INCLUDING my comments as well as the reviewer's reports and your detailed responses (as Word file).

I look forward to reading a new revised version of your manuscript as soon as possible.

Yours sincerely,

Poonam Bheda, PhD
Scientific Editor
Molecular Systems Biology

Reviewer #1:

The authors addressed my remaining points sufficiently.

I recommend including grid lines for Fig. S3 in the same way as for Fig. EV2A,B. Otherwise it is difficult to compare the degrees of correlations.

Regarding the clarification of technical vs. biological replicates, it is debatable if repeating the chemical mapping experiments with the same cell line (clone 2) amounts to a technical or biological replicate. I tend to the former. Nonetheless, I can go along with calling already this a biological replicate as long as the more stringent biological replication (comparison between the independent clones 1-2 vs. 2) was also done and is shown now in Figs. EV2G and S3.

I am still surprised that the cell lines with mutated H4 did not show a growth phenotype. I understand that they were selected for this and I agree that the mutant H4 mRNA levels of >70% suggest that the mutated H4 dominates the H4 pool. Nonetheless, the ratio of WT vs. mutant H4 on the protein level remains unclear. Nonetheless again, this is acknowledged by the authors and cannot be resolved in the present experimental design.

Reviewer #2:

The authors adequately addressed all my points.

I have no other comments.

Reviewer #3:

I am satisfied with the authors' responses to my critique.

Point-by-point response

9th Dec 2025

Manuscript Number: **MSB-2025-13024R**

Title: Differential nucleosome organization in human interphase and metaphase chromosomes

Dear Prof Wang,

Thank you for the submission of your revised manuscript to Molecular Systems Biology. I am pleased to inform you that we will be able to accept your manuscript pending the following final amendments and appropriate response to reviewers:

1) In the main manuscript file, please include keywords to max. 5.

Added in the text: Nucleosome, chromosome, mitosis, chemical mapping, DNA cyclizability.

2) Please rename "Data and availability" to "Data availability". Please also remove the reviewer access information, publicly release the datasets, and format the 'Data availability' section according to the example below.

"The datasets and computer code produced in this study are available in the following databases:

- Chip-Seq data: Gene Expression Omnibus GSE46748

(<https://www.ncbi.nlm.nih.gov/geo/query/acc.cgi?acc=GSE46748>)

- Modeling computer scripts: GitHub

(<https://github.com/SysBioChalmers/GECKO/releases/tag/v1.0>)

- [data type]: [full name of the resource] [accession number/identifier] ([doi or URL or identifiers.org/DATABASE:ACCESSION])"

Edited.

3) Please include a README file on Github with practical use instructions for potential future users of your code.

We have added a README file on Github.

<https://github.com/jipingw/numap>

4) Please rename "Competing interests" to "Disclosure and competing interests statement". We updated our journal's competing interests policy in January 2022 and request authors to consider both actual and perceived competing interests. Please review the policy <https://www.embopress.org/competing-interests> and update your competing interests if necessary.

Edited.

5) Author contributions: Please remove it from the manuscript and specify author contributions in our submission system. CRediT has replaced the traditional author contributions section because it offers a systematic machine-readable author contributions format that allows for more effective research assessment. You are encouraged to use the free text boxes beneath each contributing author's name to add specific details on the author's contribution. More information is available in our guide to authors:

<https://www.embopress.org/page/journal/17574684/authorguide#authorshipguidelines>

We deleted in the text.

6) Our journal encourages inclusion of *data citations in the reference list* to directly cite datasets that were re-used and obtained from public databases. Data citations in the article text are distinct from normal bibliographical citations and should directly link to the database records from which the data can be accessed. In the main text, data citations are formatted as follows: "Data ref: Smith et al, 2001" or "Data ref: NCBI Sequence Read Archive PRJNA342805, 2017". In the Reference list, data citations must be labeled with "[DATASET]". A data reference must provide the database name, accession number/identifiers and a resolvable link to the landing page from which the data can be accessed at the end of the reference. Further instructions are available at <https://www.embopress.org/page/journal/17574684/authorguide#referencesformat>.

Yes, we added Dataset ref.

7) In the Methods, please take care of the following:

- Cell lines: Please be sure to include a sentence in the Methods as to whether or not the cell lines were recently authenticated and tested for mycoplasma contamination. Please also be sure to update the Author Checklist with this information and where it can be found in the manuscript.

Yes, we included it.

- Please ensure that a statement on whether or not blinding was done is included in the Methods even if no blinding was done. Please also be sure to update the Author Checklist with this information and where it can be found in the manuscript.

Yes.

8) Please place individual sections of the manuscript in the following order with the correct nomenclature: Title page - Abstract & Keywords - Introduction - Results - Discussion - Methods - Data Availability - Acknowledgements - Disclosure and Competing Interests Statement - References - Figure Legends - Expanded View Figure Legends.

Reordered.

9) For the figures and figure legends, please take care of the following:

- Please make sure to update the callouts of all figures in the main manuscript text. Callouts are missing for Fig. 7E and Appendix Figures S4-S7. Callouts are also missing for the individual panels of Appendix figures. The callout for Fig. S3 should be corrected to Appendix Fig. S3.

Callouts for Fig. 7E added on page 11. Callouts for Appendix Figures S4-S7 were added on page 6.

10) Please remove the "Appendix information" section from the main manuscript.

Yes, we did.

11) In the Appendix PDF, please remove the "Appendix Supplementary Figure Legend" section and instead place the individual legends below each of the corresponding figures.

Updated accordingly.

12) Synopsis:

- Synopsis text: Please provide a separate word document including a short standfirst (maximum of 300 characters, including spaces) and up to 5 bullet points to summarise the key NEW findings. They should be designed to be complementary to the abstract - i.e. not repeat the same text. We encourage inclusion of key acronyms and quantitative information (maximum of 30 words / bullet point). Please use the passive voice.
- Please check your synopsis text and image before submission with your revised manuscript. Please be aware that in the proof stage minor corrections only are allowed (e.g., typos).

Checked and a new synopsis text file is submitted.

13) Please explain the Source Data for Figure 2B, as it is unclear how this matches the pie chart in the manuscript. In addition, the numbers are exactly the same in the columns for 'y', 'y min',

and 'y max' - please explain what these numbers represent. Please also explain the Source Data for Figure 3G, as it is also unclear what numbers are represented in the bar graph.

We updated the two source files for Fig 2B and 3G. New readme files included.

14) As part of the EMBO Publications transparent editorial process initiative (see our policy here: https://www.embopress.org/transparent-process#Review_Process), Molecular Systems Biology will publish online a Peer Review File (PRF) to accompany accepted manuscripts. This file will be published in conjunction with your paper and will include the anonymous referee reports, your point-by-point response and all pertinent correspondence relating to the manuscript. Let us know whether you agree with the publication of the PRF and as here, if you want to remove or not any figures from it prior to publication. Please note that the Authors checklist will be published at the end of the PRF.

We agree.

15) After your paper is published, we may promote it on social media. If you have any handles or hashtags for Bluesky you would like included, please let us know.

We do not use them.

16) Please provide a point-by-point letter INCLUDING my comments as well as the reviewer's reports and your detailed responses (as Word file).

Included in this note.

I look forward to reading a new revised version of your manuscript as soon as possible.

Yours sincerely,

Poonam Bheda, PhD
Scientific Editor
Molecular Systems Biology

Reviewer #1:

The authors addressed my remaining points sufficiently.
I recommend including grid lines for Fig. S3 in the same way as for Fig. EV2A,B. Otherwise it is

difficult to compare the degrees of correlations.

Regarding the clarification of technical vs. biological replicates, it is debatable if repeating the chemical mapping experiments with the same cell line (clone 2) amounts to a technical or biological replicate. I tend to the former. Nonetheless, I can go along with calling already this a biological replicate as long as the more stringent biological replication (comparison between the independent clones 1-2 vs. 2) was also done and is shown now in Figs. EV2G and S3.

I am still surprised that the cell lines with mutated H4 did not show a growth phenotype. I understand that they were selected for this and I agree that the mutant H4 mRNA levels of >70% suggest that the mutated H4 dominates the H4 pool. Nonetheless, the ratio of WT vs. mutant H4 on the protein level remains unclear. Nonetheless again, this is acknowledged by the authors and cannot be resolved in the present experimental design.

We have added grid lines for Appendix Fig. S3.

Reviewer #2:

The authors adequately addressed all my points.
I have no other comments.

Thank you.

Reviewer #3:

I am satisfied with the authors' responses to my critique.

Thank you.

13th Jan 2026

Manuscript number: MSB-2025-13024RR

Title: Differential nucleosome organization in human interphase and metaphase chromosomes

Dear Prof Wang,

Thank you again for sending us your revised manuscript. We are now satisfied with the modifications made and I am pleased to inform you that your paper has been accepted for publication.

You may qualify for financial assistance for your publication charges - either via a Springer Nature fully open access agreement or an EMBO initiative. Check your eligibility: <https://link.springer.com/journal/44320/how-to-publish-with-us>

Yours sincerely,

Poonam Bheda, PhD
Scientific Editor
Molecular Systems Biology

>>> Please note that it is Molecular Systems Biology policy for the transcript of the editorial process (containing referee reports and your response letter) to be published as an online supplement to each paper. If you do NOT want this, you will need to inform the Editorial Office via email immediately. More information is available here: <https://link.springer.com/partners/embo-press/editorial-policies#Peer%20review>